# Achieving Linear Speedup and Near-Optimal Complexity for Decentralized Optimization over Row-stochastic Networks

**Liyuan Liang** [1]  **Xinyi Chen** [1]  **Gan Luo** [1]  **Kun Yuan** [2]

## Abstract

A key challenge in decentralized optimization is determining the optimal convergence rate and designing algorithms to achieve it. While this problem has been extensively addressed for doubly-stochastic and column-stochastic mixing matrices, the row-stochastic scenario remains unexplored. This paper bridges this gap by introducing effective metrics to capture the influence of row-stochastic mixing matrices and establishing the first convergence lower bound for decentralized learning over row-stochastic networks. However, existing algorithms fail to attain this lower bound due to two key issues: deviation in the descent direction caused by the adapted gradient tracking (GT) and instability introduced by the PULL-DIAG protocol. To address descent deviation, we propose a novel analysis framework demonstrating that PULL-DIAG-GT achieves linear speedup—the first such result for row-stochastic decentralized optimization. Moreover, by incorporating a multi-step gossip (MG) protocol, we resolve the instability issue and attain the lower bound, achieving near-optimal complexity for decentralized optimization over row-stochastic networks.

## 1. Introduction

Scaling machine learning tasks to large datasets and models requires efficient distributed computing across multiple nodes. This paper investigates decentralized stochastic optimization over a network of $n$ nodes:

$$\min_{x\in\mathbb{R}^d} \quad f(x) = \frac{1}{n}\sum_{i=1}^{n}\Big[f_i(x) := \mathbb{E}_{\xi_i\sim\mathcal{D}_i}[F(x;\xi_i)]\Big] \quad (1)$$

where $\xi_i$ is a random data vector supported on $\Xi_i \subseteq \mathbb{R}^q$ with some distribution $\mathcal{D}_i$, and $F : \mathbb{R}^d \times \mathbb{R}^q \to \mathbb{R}$ is a Borel-measurable function. Each loss function $f_i$ is accessible only by node $i$ and is assumed to be smooth and potentially nonconvex. Note that data heterogeneity typically exists, i.e., the local data distributions $\{\mathcal{D}_i\}_{i=1}^{n}$ vary across nodes. In this paper, we model decentralized communication between nodes as a *directed graph*, a scenario frequently encountered in practical applications. For example, bidirectional communication may be infeasible due to differences in node power ranges (Yang et al., 2019) or connection failures (Yemini et al., 2022; Li et al., 2024). In distributed deep learning, carefully designed directed topologies can achieve sparser and faster communication than their undirected counterparts, thereby reducing the training wall clock time (Bottou et al., 2018; Assran et al., 2019; Yuan et al., 2021).

**Network topology and mixing matrix.** A central challenge in decentralized optimization is determining the optimal convergence rate and designing algorithms that achieve it. This requires a theoretical understanding of how network topologies influence decentralized algorithms. For any given connected network, its topology can be represented by a mixing matrix that reflects its connectivity pattern, serving as a critical tool for evaluating the network's impact. In undirected networks, symmetric and doubly-stochastic matrices can be readily constructed. However, in directed networks, constructing a doubly-stochastic mixing matrix is generally infeasible. Instead, mixing matrices are typically either column-stochastic (Nedić & Olshevsky, 2014; Nedić et al., 2017) or row-stochastic (Sayed, 2014; Mai & Abed, 2016), but not both.

**Optimal complexity over doubly-stochastic networks is well-established.** The connectivity of a doubly-stochastic mixing matrix can be effectively assessed using the spectral gap, a metric that quantifies how closely a decentralized network approximates a fully connected one. Leveraging this metric, several studies have established the optimal convergence rates for decentralized algorithms. For example,

---

[1]School of Mathematics Science, Peking University, Beijing, China  [2]Center for Machine Learning Research, Peking University, Beijing, China. Correspondence to: Kun Yuan <kunyuan@pku.edu.cn>.

*Proceedings of the 42$^{nd}$ International Conference on Machine Learning*, Vancouver, Canada. PMLR 267, 2025. Copyright 2025 by the author(s).

references (Scaman et al., 2017; 2018; Sun & Hong, 2019; Kovalev et al., 2021) derive optimal convergence rates for convex or non-stochastic decentralized optimization. Lu & De Sa (2021) determine the optimal complexity for non-convex and stochastic decentralized optimization over a specific type of linear network, while Yuan et al. (2022) extend this complexity to a significantly broader class of networks. The optimal complexity for doubly-stochastic time-varying networks has been established in (Huang & Yuan, 2022; Li & Lin, 2024; Kovalev et al., 2021).

**Optimal complexity over column-stochastic networks is established recently.** If out-degree information is available prior to communication, a column-stochastic matrix can be readily constructed. When decentralized algorithms rely solely on column-stochastic matrices, this is referred to as the COL-ONLY setting. The PUSH-SUM gossip protocol (Kempe et al., 2003; Tsianos et al., 2012) forms the foundation of COL-ONLY algorithms. Many algorithms based on PUSH-SUM achieve superior convergence rates, e.g., Nedić & Olshevsky (2015); Tsianos et al. (2012); Zeng & Yin (2017); Xi & Khan (2017); Xi et al. (2017); Nedić et al. (2017); Assran et al. (2019); Qureshi et al. (2020). However, these works fail to precisely capture the influence of column-stochastic networks and, as a result, do not clarify the optimal complexity in the COL-ONLY setting. This open question is addressed in a recent study by Liang et al. (2023), which introduces effective metrics to evaluate the influence of column-stochastic networks, establishes the optimal lower bound for the COL-ONLY setting, and proposes algorithms that achieve this bound.

**Optimal complexity over row-stochastic networks remains unclear yet.** If out-degree information is unavailable, column-stochastic matrices cannot be directly constructed. However, row-stochastic matrices can be formed using in-degree information, which can be easily obtained by counting received messages. This is referred to as the ROW-ONLY setting. Similar to how PUSH-SUM serves as the basis for COL-ONLY algorithms, the foundation of ROW-ONLY methods is the PULL-DIAG gossip protocol (Mai & Abed, 2016). Building on PULL-DIAG, Mai & Abed (2016) adapted the distributed gradient descent (DGD) algorithm for the ROW-ONLY setting, while Li et al. (2019); Xin et al. (2019c) extended gradient tracking methods, and Ghaderyan et al. (2023); Lü et al. (2020); Xin et al. (2019a) introduced momentum-based ROW-ONLY gradient tracking. However, the convergence analysis for ROW-ONLY algorithms is still quite limited. Current analyses focus only on deterministic and strongly convex loss functions, leaving the performance of ROW-ONLY algorithms in non-convex and stochastic settings unknown. More importantly, the impact of row-stochastic networks on the convergence rate of ROW-ONLY algorithms remains unclear. These gaps present significant obstacles to determining the optimal complexity in the ROW-ONLY setting. Some fundamental open problems are:

Q1. What are the effective metrics that can fully capture the impact of row-stochastic networks on decentralized stochastic optimization, and how do they influence the convergence of prevalent ROW-ONLY algorithms?

Q2. Given these metrics, what is the lower bound on the convergence rate for ROW-ONLY algorithms in the non-convex and stochastic setting?

Q3. Can existing ROW-ONLY algorithms readily achieve the optimal convergence rate? If not, what limitations do they face?

Q4. Can we develop new ROW-ONLY algorithms that overcome the limitations of existing algorithms and attain the aforementioned lower bound?

**Main contributions.** This paper improves the understanding of decentralized methods over row-stochastic networks by addressing these open questions. Our contributions are :

C1. We find that the metrics *generalized spectral gap* and *equilibrium skewness*, proposed by (Liang et al., 2023) to characterize the influence of column-stochastic networks, can also effectively capture the impact of row-stochastic networks on decentralized algorithms.

C2. Using these metrics, we establish the *first* lower bound on the convergence rate for nonconvex decentralized stochastic first-order algorithms with a row-stochastic mixing matrix. This bound achieves linear speedup with respect to network size $n$ and captures the influence of gradient noise, the mixing matrix, the number of nodes and the smoothness of the loss function.

C3. We find existing ROW-ONLY algorithms cannot attain the aforementioned lower bound due to two challenges. First, the use of row-stochastic mixing matrices alone introduces a deviation in the descent direction from the globally averaged gradient, preventing existing analyses from achieving linear speedup convergence. Second, the PULL-DIAG protocol introduces inversion of small values during operation, which introduces instability in ROW-ONLY algorithms.

C4. We develop a novel analysis framework proving that PULL-DIAG-GT achieves linear speedup, marking the *first* such result in ROW-ONLY scenarios. Moreover, when integrated with a multistep gossip (MG) protocol, MG-PULL-DIAG-GT addresses the instability caused by the inversion of small values and achieves the established lower bound. Therefore, both the lower bound and our algorithm achieve optimal complexity for decentralized learning over row-stochastic networks.

**Notations.** Let $\mathbb{1}_n$ denote the vector of all-ones of $n$ dimensions and $I_n \in \mathbb{R}^{n \times n}$ the identity matrix. We let matrix $A$ denote the row-stochastic matrix ($A\mathbb{1}_n = \mathbb{1}_n$). The set $[n]$ represents the indices $\{1, 2, \ldots, n\}$. $\mathrm{Diag}(A)$ refers to the diagonal matrix composed of the diagonal entries of $A$, while $\mathrm{diag}(v)$ is the diagonal matrix derived from vector $v$. The Perron vector of matrix $A$ is $\pi_A$, i.e., $\pi_A \geq 0, \pi_A^\top A = \pi_A^\top, \pi_A^\top \mathbb{1} = 1$. By letting $\Pi_A := \mathrm{diag}(\pi_A)$, we define $\|v\|_{\pi_A} := \|\Pi_A^{1/2} v\|$, which is associated with the induced matrix norm $\|W\|_{\pi_A} := \|\Pi_A^{1/2} W \Pi_A^{-1/2}\|_2$. We define $A_\infty := \mathbb{1}_n \pi_A^\top$. The vector $\boldsymbol{x}_i^{(k)} \in \mathbb{R}^d$ represents the local model at node $i$ during iteration $k$. We also define $n \times d$ matrices

$$\mathbf{x}^{(k)} := [(\boldsymbol{x}_1^{(k)})^\top; (\boldsymbol{x}_2^{(k)})^\top; \cdots; (\boldsymbol{x}_n^{(k)})^\top]$$

$$\nabla F(\mathbf{x}^{(k)}; \boldsymbol{\xi}^{(k)}) := [\nabla F_1(\boldsymbol{x}_1^{(k)}; \xi_1^{(k)})^\top; \cdots; \nabla F_n(\boldsymbol{x}_n^{(k)}; \xi_n^{(k)})^\top]$$

$$\nabla f(\mathbf{x}^{(k)}) := [\nabla f_1(\boldsymbol{x}_1^{(k)})^\top; \cdots; \nabla f_n(\boldsymbol{x}_n^{(k)})^\top]$$

by stacking all local variables. The upright bold symbols (e.g. $\mathbf{x}, \mathbf{w}, \mathbf{g}$) always denote network-level quantities. We define filtration $\mathcal{F}_k$ as the collection of all the information available up to $\mathbf{x}^{(k)}$, excluding the stochastic gradient evaluated at $\mathbf{x}^{(k)}$. We use the symbol $\lesssim$ to represent inequality up to absolute constants.

## 2. Metrics for Row-stochastic Networks

We consider a directed network with $n$ computing nodes that is associated with a mixing matrix $A = [a_{ij}]_{i,j=1}^n \in \mathbb{R}^{n \times n}$ where $a_{ij} \in (0, 1]$ if node $j$ can send information to node $i$ otherwise $a_{ij} = 0$. Decentralized optimization is built upon partial averaging $\boldsymbol{z}_i^+ = \sum_{j \in \mathcal{N}_i} a_{ij} \boldsymbol{z}_j$ in which $\boldsymbol{z}_i \in \mathbb{R}^d$ is a local vector held by node $i$ and $\mathcal{N}_i$ denotes the in-neighbors of node $i$, including node $i$ itself. Since every node conducts partial averaging simultaneously, we have

$$\mathbf{z} \triangleq [\boldsymbol{z}_1^\top; \boldsymbol{z}_2^\top; \cdots; \boldsymbol{z}_n^\top] \xmapsto{\text{A-protocol}} \mathbf{z}^+ = A\mathbf{z} \qquad (2)$$

where $A$-protocol represents partial averaging with mixing matrix $A$. Evidently, the algebraic characteristics of $A$ substantially affect the convergence of partial averaging and the corresponding decentralized optimization. This section explores metrics that capture the characteristics of $A$.

### 2.1. Row-stochastic Mixing Matrix

This paper focuses on a static directed network $\mathcal{G} = (V, \mathcal{E})$ associated with a row-stochastic matrix $A = [a_{ij}]_{n \times n}$.

**Assumption 1** (Primitive and Row-stochastic). *The mixing matrix $A$ is nonnegative, primitive and satisfies $A\mathbb{1}_n = \mathbb{1}_n$. The weight $a_{ij} \in (0, 1]$, if $(i \to j) \in \mathcal{E}$, otherwise $a_{ij} = 0$.*

If $\mathcal{G}$ is strongly-connected, i.e., there exists a directed path from each node to every other node, and $A$ has a positive

trace, then $A$ is primitive. It is straightforward to make $A$ row-stochastic by setting $a_{ij} = 1/(1 + d_i^{\mathrm{in}})$ if $(i, j) \in \mathcal{E}$ or $j = i$ otherwise $a_{ij} = 0$, where $\mathcal{E}$ is the set of directed edges and $d_i^{\mathrm{in}}$ is the in-degree of node $i$ excluding the self-loop. With Assumption 1, we have the following result:

**Proposition 1** (Perron-Frobenius theorem (Perron, 1907)). *If matrix $A$ satisfies Assumption 1, there exists a unique equilibrium vector $\pi_A \in \mathbb{R}^n$ with positive entries such that*

$$\pi_A^\top A = \pi_A^\top, \quad \mathbb{1}_n^\top \pi_A = 1, \quad \text{and} \quad \lim_{k \to \infty} A^k = \mathbb{1}_n \pi_A^\top.$$

### 2.2. Effective Metrics to Characterize Row-stochastic $A$

Most decentralized algorithms rely on gossip protocols like (2), where local variables are partially mixed to approximate the global average. The properties of the mixing matrix $A$ play a critical role in determining both the feasibility of achieving the global average and the efficiency of this process. These properties serve as key metrics for assessing the influence of $A$ on algorithmic performance.

Now we examine the $A$-protocol (2) where $A$ is a row-stochastic mixing matrix $A$. Suppose that each node $i$ has a local variable $\boldsymbol{z}_i \in \mathbb{R}^d$, we let $\mathbf{z} = [\boldsymbol{z}_1^\top; \boldsymbol{z}_2^\top; \cdots; \boldsymbol{z}_n^\top] \in \mathbb{R}^{n \times d}$ and initialize $\mathbf{x}^{(0)} = \mathbf{z}$. Following the gossip protocol as in (2), we have the following recursions:

$$\mathbf{x}^{(k)} = A\mathbf{x}^{(k-1)} = A^k \mathbf{x}^{(0)} \xmapsto{k \to \infty} \mathbb{1}_n \pi_A^\top \mathbf{z}, \qquad (3)$$

where we utilize the property $\lim_{k \to \infty} A^k = \mathbb{1}_n \pi_A^\top$ (see Proposition 1). It is evident that the matrix $A$ influences both whether and how quickly $\mathbf{x}^{(k)}$ approaches the global average $\mathbb{1}_n \mathbb{1}_n^\top \mathbf{z}/n$. Inspired by (3), we use the following two metrics to characterize the row-stochastic matrix $A$:

- The **generalized spectral gap** $1 - \beta_A$ of the row-stochastic matrix $A$, where

$$\beta_A := \left\| A - \mathbb{1}_n \pi_A^\top \right\|_{\pi_A} = \| A - A_\infty \|_{\pi_A} \in [0, 1)$$

quantifies the convergence rate of $\mathbf{x}^{(k)}$ to the weighted average $\mathbb{1}_n \pi_A^\top \mathbf{z}$ in (3). The $\pi_A$-norm was defined in the notations in Sec. 1. As $\beta_A$ approaches 0, the iterates $\mathbf{x}^{(k)}$ converge more rapidly to the fixed point $\mathbb{1}_n \pi_A^\top \mathbf{z}$ of the $A$-protocol (3).

- The **equilibrium skewness**

$$\kappa_A := \max(\pi_A)/\min(\pi_A) \in [1, +\infty)$$

captures the disagreement between the equilibrium vector $\pi_A$ and the uniform vector $n^{-1}\mathbb{1}_n$. When $\kappa_A \to 1$, it holds that $\pi_A \to \mathbb{1}_n/n$, and hence, the weighted average aligns better with the global average $\mathbb{1}_n \mathbb{1}_n^\top \mathbf{z}/n$.

The spectral gap gauges the rate of the $A$-protocol when converging to the fixed point $\mathbb{1}_n \pi_A^\top \mathbf{z}$, while equilibrium skewness measures the proximity of the fixed point to the desired global average $\mathbb{1}_n \mathbb{1}_n^\top \mathbf{z}/n$. Together, these metrics effectively capture the influence of row-stochastic mixing matrices and directed networks on decentralized algorithms. Omitting either would lead to an incomplete understanding of their impact. Figure 1 shows examples that the spectral gap and equilibrium skewness jointly impact the convergence to average consensus. The protocol used in Figure 1 is called PULL-DIAG and will be discussed in Sec. 2.3.

It is important to note that these two metrics are not new; they were proposed in (Xin et al., 2019b; Liang et al., 2023) to assess the influence of column-stochastic mixing matrices. Our contribution lies in demonstrating that these metrics are also applicable to row-stochastic mixing matrices. Prior to our work, no literature had examined the metrics that can gauge the influence of row-stochastic mixing matrices on decentralized algorithms.

## 2.3. Pull-Diag Protocol Corrects Weighted Average

According to (3), $A$-protocol alone cannot achieve the desired global average. The bias between the limiting weighted average $\mathbb{1}_n \pi_A^\top \mathbf{z}$ and the desired global average $\mathbb{1}_n \mathbb{1}_n^\top \mathbf{z}/n$ can be corrected by the following manner:

$$A^k \mathrm{diag}(n\pi_A)^{-1} \mathbf{z}$$
$$\overset{k\to\infty}{\longmapsto} \mathbb{1}_n \pi_A^\top \mathrm{diag}(n\pi_A)^{-1} \mathbf{z} = \mathbb{1}_n \mathbb{1}_n^\top \mathbf{z}/n. \quad (4)$$

Although the above strategy is effective, the quantity $\pi_A$ is not known beforehand. To estimate $\pi_A$, prior works (Mai & Abed, 2016; Xin et al., 2019c; Ghaderyan et al., 2023) use power iterations, resulting in a practical and efficient approach referred to as PULL-DIAG in this paper:

$$V_{k+1} = AV_k, \quad D_{k+1} = \mathrm{Diag}(nV_{k+1}), \quad (5a)$$
$$\mathbf{z}^{(k+1)} = V_{k+1} D_{k+1}^{-1} \mathbf{z}. \quad (5b)$$

With initialization $V_0 = I_n$, we have $V_k = A^k$ and $D_k = \mathrm{Diag}(nA^k)$. It holds that $V_k \to \mathbb{1}_n \pi_A^\top$ and $D_k \to \mathrm{diag}(n\pi_A)$ as $k \to \infty$. Substituting these facts into (5b), we asymptotically achieve the bias correction illustrated in (4). The distributed implementation details of the PULL-DIAG protocol can be found in Appendix B. It is shown in Figure 1 that PULL-DIAG protocol corrects the weighted average and converges exponentially fast.

# 3. Convergence Lower Bounds

## 3.1. Assumptions

This subsection specifies the category of decentralized algorithms to which our lower bound applies.

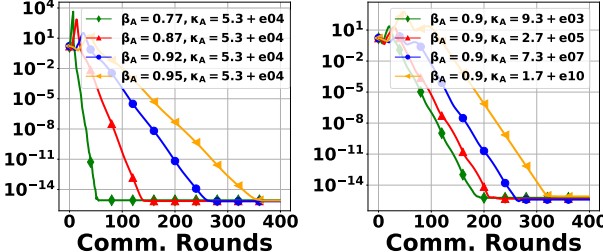

*Figure 1.* Convergence of PULL-DIAG protocol on different mixing matrices with varying spectral gaps ($\beta_A$) and equilibrium skewness ($\kappa_A$). The $y$-axis represents the consensus error $\|\mathbf{z}^{(k)} - \mathbb{1}_n \mathbb{1}_n^\top \mathbf{z}/n\|$. The left plot shows fixed $\kappa_A$ with different $\beta_A$, while the right plot shows fixed $\beta_A$ with different $\kappa_A$.

**Function class.** We define the function class $\mathcal{F}_{\Delta,L}$ as the set of functions that satisfy Assumption 2, for any given dimension $d \in \mathbb{N}_+$ and any initialization point $x^{(0)} \in \mathbb{R}^d$.

**Assumption 2** (Smoothness). *There exists constants* $L, \Delta \geq 0$ *such that*

$$\|\nabla f_i(\boldsymbol{x}) - \nabla f_i(\boldsymbol{y})\| \leq L\|\boldsymbol{x} - \boldsymbol{y}\|, \forall i \in n, \boldsymbol{x}, \boldsymbol{y} \in \mathbb{R}^d.$$

*and for initial model parameter* $\boldsymbol{x}^{(0)}$,

$$f_i(\boldsymbol{x}^{(0)}) - \inf_{\boldsymbol{x}\in\mathbb{R}^d} f_i(\boldsymbol{x}) \leq \Delta, \quad \forall i \in [n].$$

**Gradient oracle class.** We assume that each node $i$ processes its local cost function $f_i$ using a stochastic gradient oracle $\nabla F(\boldsymbol{x}; \xi_i)$, which provides unbiased estimates of the exact gradient $\nabla f_i$ with bounded variance. Specifically, we define the stochastic gradient oracle class $O_{\sigma^2}$ as the set of all oracles $\nabla F(\cdot; \xi_i)$ that satisfy Assumption 3.

**Assumption 3** (Gradient oracles). *There exists a constant* $\sigma \geq 0$ *such that for all* $\boldsymbol{x} \in \mathbb{R}^d$, $i \in [n]$ *we have*

$$\mathbb{E}[\nabla F_i(\boldsymbol{x}; \xi)] = \nabla f_i(\boldsymbol{x}), \quad \mathrm{tr}(\mathrm{Var}[\nabla F_i(\boldsymbol{x}; \xi)]) \leq \sigma^2.$$

*We also assume that the gradient noise is linearly independent, i.e.,* $\forall i \neq j \in [n]$ *we have*

$$\mathrm{Cov}(\nabla F_i(\boldsymbol{x}_i; \xi_i), \nabla F_j(\boldsymbol{x}_j; \xi_j)) = 0.$$

**Algorithm class description.** We focus on decentralized algorithms where each node $i$ maintains a local solution $\boldsymbol{x}_i^{(k)}$ at iteration $k$ and communicates using the $A$-protocol defined in (2). These algorithms also adhere to the linear-spanning property, as defined in (Carmon et al., 2020; 2021; Yuan et al., 2022; Lu & De Sa, 2021). Informally, this property ensures that each local solution $\boldsymbol{x}_i^{(k)}$ resides within the linear space spanned by $\boldsymbol{x}_i^{(0)}$, its local stochastic gradients, and interactions with neighboring nodes. Upon completion of $K$ iterations, the final output $\hat{\boldsymbol{x}}^{(K)}$ can be any variable in $\mathrm{span}(\{\{\boldsymbol{x}_i^{(k)}\}_{i=1}^n\}_{k=0}^K)$. Let $\mathcal{A}_A$ denote the set of all algorithms that adhere to partial averaging via mixing matrix $A$ and satisfy the linear-spanning property.

## 3.2. Lower Bound

With $\beta_A$ and $\kappa_A$ at hand, we show, for the first time, that the convergence rate of any non-convex decentralized stochastic first-order algorithm with a row-stochastic mixing matrix is lower-bounded by the following theorem.

---

**Theorem 1** (Lower bound). *For any given $L \geq 0$, $n \geq 2$, $\sigma \geq 0$, and $\tilde{\beta} \in [0.01, 1 - 1/n]$, there exists a set of loss functions $\{f_i\}_{i=1}^n \in \mathcal{F}_{\Delta, L}$, a set of stochastic gradient oracles in $\mathcal{O}_{\sigma^2}$, and a row-stochastic matrix $A \in \mathbb{R}^{n \times n}$ with $\beta_A = \tilde{\beta}$ and $\ln(\kappa_A) = \Omega(n(1 - \beta_A))$, such that the convergence of any algorithm $\mathscr{A} \in \mathcal{A}_A$ starting from $x_i^{(0)} = x^{(0)}, i \in [n]$ with $K$ iterations is lower bounded by*

$$\mathbb{E}\|\nabla f(\hat{x}^{(K)})\|^2 = \Omega\left(\frac{\sigma\sqrt{L\Delta}}{\sqrt{nK}} + \frac{(1 + \ln(\kappa_A))L\Delta}{(1 - \beta_A)K}\right), \tag{6}$$

*where $K$, $\sigma$, $L$, and $\Delta$ represent the total number of iterations, the gradient variance, the smoothness parameter of the functions, and the initial gap in the function values, respectively. The proof is in Appendix A.*

---

**Linear speedup.** The first term, $\sigma/\sqrt{nK}$, dominates the lower bound (6) when $K$ is sufficiently large, indicating that decentralized algorithms with row-stochastic mixing matrices could achieve linear speedup with respect to network size $n$ (*i.e.*, convergence improves as the number of computing nodes $n$ increases).

**Network topology impact.** The lower bound in (6) explicitly highlights the combined impact of the generalized spectral gap $\beta_A$ and the equilibrium skewness $\kappa_A$ on decentralized algorithms utilizing row-stochastic mixing matrices. Omitting either metric would provide an incomplete understanding of the algorithmic performance.

**Deterministic scenario.** When the gradient noise $\sigma = 0$, the established lower bound in (6) for stochastic settings simplifies to the first lower bound for deterministic decentralized algorithms with row-stochastic mixing matrices.

## 4. Achieving Linear Speedup Using Row-stochastic Matrix Alone

The lower bound (6) reveals a linear speedup convergence rate of $\sigma/\sqrt{nK}$ as $K$ grows large. However, no existing ROW-ONLY decentralized stochastic algorithm has theoretically achieved this rate, highlighting a significant gap from the lower bound. This section identifies the challenges and presents a novel analysis framework that achieves the first theoretical linear speedup for ROW-ONLY algorithms.

## 4.1. Pull-DIAG-GT Algorithm

We begin by reviewing the state-of-the-art ROW-ONLY algorithms (Li et al., 2019; Xin et al., 2019c; Ghaderyan et al., 2023; Lü et al., 2020; Xin et al., 2019a), all of which are based on PULL-DIAG-GT—an adaptation of gradient tracking (Nedic et al., 2017; Di Lorenzo & Scutari, 2016; Xu et al., 2015; Qu & Li, 2017) designed specifically for ROW-ONLY scenarios.

$$x^{(k+1)} = A(x^{(k)} - \alpha y^{(k)}), \tag{7a}$$

$$y^{(k+1)} = A(y^{(k)} + D_{k+1}^{-1} g^{(k+1)} - D_k^{-1} g^{(k)}). \tag{7b}$$

Here, $D_k = \text{Diag}(A^k), \forall k \geq 1$, $D_0 = I_n$. Matrix $x^{(k)}$ denotes the stacked model parameters and $y^{(k)}$ denotes the gradient tracking term. $g^{(k)}$ denotes the stochastic gradient, defined as $g^{(k)} = \nabla F(x^{(k)}; \xi^{(k)})$, with $y^{(0)} = g^{(0)}$. The details of algorithm implementation is provided in Appendix B. As recursion (7) incorporates the PULL-DIAG protocol (5) into gradient tracking, it is termed PULL-DIAG-GT throughout this paper.

It is noteworthy $D_k^{-1}$ may involve inversion of zeros. Therefore, the following assumption is necessary:

**Assumption 4** (Bounded Diagonals). *There exists a constant $\theta_A > 0$ such that*

$$[A^k]_{ii}^{-1} \leq \theta_A, \ \forall k \geq 1, \ i \in [n].$$

*Remarkably, under Assumption 1, the existence of $\theta_A$ can be guaranteed if and only if every node has a self-loop.*

**Algorithm insight.** As shown in $A$-protocol (3), communication utilizing a row stochastic matrix yields a biased average $z \to \pi_A^\top z$. In PULL-DIAG-GT, we can left-multiply $\pi_A^\top$ on both sides of (7a) and observe the following dynamics:

$$\pi_A^\top x^{(k+1)} = \pi_A^\top x^{(k)} - \alpha \pi_A^\top y^{(k)}. \tag{8}$$

If $y^{(k)}$ represents the stacked stochastic gradients, i.e., $y^{(k)} = \nabla F(x^{(k)}; \xi^{(k)})$, the descent direction $\pi_A^\top y^{(k)}$ deviates from the desired globally averaged gradient $\mathbb{1}_n^\top y^{(k)}/n$, preventing $\pi_A^\top x^{(k)}$ from converging to the solution to problem (1). To ensure convergence, PULL-DIAG-GT corrects the descent direction using $D_k^{-1}$ in (7b). Left-multiplying $\pi_A^\top$ on both sides of (7b) yields

$$\pi_A^\top y^{(k+1)} - \pi_A^\top D_{k+1}^{-1} g^{(k+1)} = \pi_A^\top y^{(k)} - \pi_A^\top D_k^{-1} g^{(k)}$$

$$= \cdots = \pi_A^\top y^{(0)} - \pi_A^\top g^{(0)} \overset{(a)}{=} 0,$$

where equality (a) holds due to $y^{(0)} = g^{(0)}$. This implies

$$\pi_A^\top y^{(k)} = \pi_A^T D_k^{-1} g^{(k)} \overset{(b)}{\approx} \mathbb{1}_n^\top g^{(k)} \triangleq n\bar{g}^{(k)}, \tag{9}$$

where (b) holds because $D_k = \text{Diag}(A^k) \approx \text{diag}(\pi_A)$, and $\bar{g}^{(k)} = (1/n)\mathbb{1}_n^\top g^{(k)}$. The combined dynamics of (8)

and (9) ensure that PULL-DIAG-GT converges along the globally averaged gradient, ultimately solving problem (1).

**Challenges in establishing linear speedup.** Although the rationale behind PULL-DIAG-GT's convergence to the desired solution is clear, to the best of our knowledge, no existing analysis has established its linear speedup convergence with respect to the network size $n$. This subsection highlights the key challenges involved.

We first define two error terms to facilitate analysis:

- **Consensus error.** Let $\pi_A^\top \mathbf{x}^{(k)}$ be the centroid variable for $\mathbf{x}^{(k)}$. We introduce $\|\mathbf{x}^{(k)} - \mathbb{1}_n \pi_A^\top \mathbf{x}^{(k)}\|^2$ as the consensus error to gauge the difference between each local variable $\boldsymbol{x}_i^{(k)}$ to the centroid $\pi_A^\top \mathbf{x}^{(k)}$.

- **Descent deviation.** We define $\|\pi_A^\top \mathbf{y}^{(k)} - n\bar{\boldsymbol{g}}^{(k)}\|^2$ as the descent deviation, measuring the discrepancy between the PULL-DIAG-GT descent direction and the desired globally averaged stochastic gradient.

If the aforementioned two error terms are guaranteed to diminish to zero, PULL-DIAG-GT enables each $\boldsymbol{x}_i^{(k)}$ to converge to centroid $\boldsymbol{w}^{(k)} = \pi_A^\top \mathbf{x}^{(k)}$, and the update of $\boldsymbol{w}^{(k)}$ asymptotically approximate centralized parallel SGD:

$$\boldsymbol{w}^{(k+1)} = \boldsymbol{w}^{(k)} - n\alpha \bar{\boldsymbol{g}}_w^{(k)}, \tag{10}$$

where $\bar{\boldsymbol{g}}_w^{(k)} = (1/n)\sum_{i=1}^n \nabla F(\boldsymbol{w}^{(k)}; \boldsymbol{\xi}_i^{(k)})$ is the globally averaged gradient over centroid. This ensures $\boldsymbol{w}^{(k)}$ to converge to the desired solution to problem (1).

Linear speedup analysis in distributed stochastic optimization relies heavily on the descent structure aligned with the globally averaged stochastic gradient (Yu et al., 2019; Xin et al., 2019b; Koloskova et al., 2020; Yang et al., 2021). When each local stochastic gradient $\boldsymbol{g}_i$ introduces mean-square gradient noise $\sigma^2$, the globally averaged gradient $\bar{\boldsymbol{g}}$ benefits from reduced noise $\sigma^2/n$, forming the foundation for linear speedup. Gradient tracking with doubly or column-stochastic mixing matrices preserves the globally averaged gradient descent direction $\bar{\boldsymbol{g}}$, enabling well-established linear speedup convergence (Koloskova et al., 2020; Lu & De Sa, 2021; Assran et al., 2019; Kungurtsev et al., 2023; Liang et al., 2023). In contrast, PULL-DIAG-GT suffers from an additional descent deviation between $\pi_A^\top \mathbf{y}$ and $\bar{\boldsymbol{g}}$, making existing linear speedup analyses inapplicable and necessitating new techniques to address this limitation.

### 4.2. Achieving linear speedup in Pull-Diag-GT

This subsection presents a new analytical framework to establish the linear speedup rate for PULL-DIAG-GT.

**Descent lemma.** Our analysis begins with a descent lemma.

**Lemma 2** (Descent lemma). *Under Assumptions 1-4, when* $\alpha \le \frac{1}{2nL}$, *for any* $k \ge 0$ *we have*

$$\frac{n\alpha}{2}\|\nabla f(\boldsymbol{w}^{(k)})\|^2 \le f(\boldsymbol{w}^{(k)}) - \mathbb{E}[f(\boldsymbol{w}^{(k+1)})|\mathcal{F}_k]$$
$$+ \alpha L^2 \underbrace{\|\Delta_x^{(k)}\|_F^2}_{\text{consensus error}} + \frac{\alpha}{n} \underbrace{\|\mathbb{E}[\pi_A^\top \mathbf{y}^{(k)} - n\bar{\boldsymbol{g}}^{(k)}|\mathcal{F}_k]\|^2}_{\text{descent deviation}}$$
$$+ \frac{\alpha^2 L\sigma^2}{2}d_k - \frac{\alpha}{4n}\|\pi_A^\top D_k^{-1}\nabla f(\mathbf{x}^{(k)})\|^2 \tag{11}$$

*where* $\boldsymbol{w}^{(k)} := \pi_A^\top \mathbf{x}^{(k)}$ *is the centroid variable,* $d_k := \sum_{j=1}^n (\frac{[\pi_A]_j}{[D_k]_j})^2$ *is a constant.* $\Delta_x^{(k)} := \mathbf{x}^{(k)} - \mathbb{1}_n \pi_A^\top \mathbf{x}^{(k)}$, *and* $\bar{\boldsymbol{g}}^{(k)} = (1/n)\mathbb{1}_n^\top \nabla F(\mathbf{x}^{(k)}; \boldsymbol{\xi}^{(k)})$.

As anticipated, the descent lemma incorporates both the consensus error and the descent deviation. Due to the conditional expectation operation, the descent deviation in the lemma appears as $\|\mathbb{E}[\pi_A^\top \mathbf{y}^{(k)} - n\bar{\boldsymbol{g}}^{(k)} \mid \mathcal{F}_k]\|^2$. In contrast, the descent lemma for gradient tracking using doubly or column-stochastic mixing matrices ensures $\bar{\boldsymbol{y}}^{(k)} = \bar{\boldsymbol{g}}^{(k)}$, thereby accounting solely for the consensus error.

**Estimate descent deviation.** According to (9), we have $\pi_A^\top \mathbf{y}^{(k)} = \pi_A^T D_k^{-1} \mathbf{g}^{(k)}$. This implies that

$$\|\mathbb{E}[\pi_A^\top \mathbf{y}^{(k)} - n\bar{\boldsymbol{g}}^{(k)}|\mathcal{F}_k]\| = \|(\pi_A^\top D_k^{-1} - \mathbb{1}_n^\top)\nabla f(\mathbf{x}^{(k)})\|.$$

Therefore, we can use the following lemma to provide an estimate for the descent deviation.

**Lemma 3.** *Under Assumptions 1, 2 and 4, for all* $k \ge 1$,

$$\|(\pi_A^\top D_k^{-1} - \mathbb{1}_n^\top)\nabla f(\mathbf{x}^{(k)})\|^2$$
$$\le 2n\kappa_A \theta_A^2 \beta_A^{2k}\left(L^2\|\Delta_x^{(k)}\|_F^2 + 2nL((f(\boldsymbol{w}^{(k)}) - f^*))\right),$$

*where* $f^* := n^{-1}\sum_{i=1}^n f_i^*$.

Lemma 3 employs $f(\boldsymbol{w}^{(k)}) - f^*$ to establish an upper bound on the norm of the stacked gradients. In many instances, this approach can negatively impact the recursive nature of the descent lemma. Nevertheless, in this particular situation, we can effectively accommodate it due to its $\mathcal{O}(\beta_A^{2k})$ coefficient. Details are provided in Lemma 13 in Appendix C.

**Estimate consensus error.** We present a novel method to establish bounds on the consensus error, which is based on converting $\Delta_x$ into rolling sums.

**Lemma 4** (Informal). *Denote* $\Delta_y^{(k)} = \mathbf{y}^{(k)} - \mathbb{1}_n \pi_A^\top \mathbf{y}^{(k)}$, $\Delta_g^{(-1)} = \mathbf{g}^{(0)}, \Delta_g^{(i)} = \mathbf{g}^{(i+1)} - \mathbf{g}^{(i)}, \forall i \ge 0$. *For any* $k \ge 0$, *it follows that*

$$\Delta_x^{(k+1)} = -\alpha \sum_{i=0}^k (A - A_\infty)^{k+1-i}\Delta_y^{(i)},$$

$$\Delta_y^{(k+1)} = \sum_{i=-1}^k \left((A - A_\infty)^{k+1-i}D_{i+1}^{-1} + \mathcal{O}(k\beta_A^k)\right)\Delta_g^{(i)}.$$

The following lemma tells us how to estimate rolling sums:

**Lemma 5** (Rolling sum). *For $A \in \mathbb{R}^{n \times n}$ satisfying Assumption 1, the following estimation holds for any matrices $\Delta^{(i)} \in \mathbb{R}^{n \times d}$ and for any $K \geq 0$:*

$$\sum_{k=0}^{K} \| \sum_{i=0}^{k} (A - A_\infty)^{k+1-i} \Delta^{(i)} \|_F^2 \leq s_A^2 \sum_{i=0}^{K} \| \Delta^{(i)} \|_F^2,$$

*where $s_A$ is a constant decided by $A$.*

Under the $L$-smooth assumption, it is straightforward to derive an upper bound for the sum $\sum_{k=0}^{K} \| \Delta_g^{(k)} \|_F^2$. Following the sequence of steps $\Delta_g \to \Delta_y \to \Delta_x$, Lemmas 4 and 5 together lead to the following consensus lemma.

**Lemma 6** (Consensus lemma, informal). *With Assumptions 1, 2, 3 and 4, we have*

$$\sum_{k=0}^{K} \mathbb{E}[\| \Delta_x^{(k+1)} \|_F^2] \leq C_{x,y} \alpha^4 \sum_{k=0}^{K} \mathbb{E}[\| \pi_A^\top D_k^{-1} \nabla f(\mathbf{x}^{(k)}) \|_F^2]$$
$$+ C_{x,0} \alpha^2 \| \nabla f(\boldsymbol{x}^{(0)}) \|^2 + C_{x,\sigma} \alpha^2 (K+1) \sigma^2,$$

*where $C_{x,y}, C_{x,0}$ and $C_{x,\sigma}$ are constants.*

**Achieving linear speedup.** Building on Lemmas 2, 3 and 6, we finally achieve the convergence Theorem 2.

> **Theorem 2** (PULL-DIAG-GT convergence). *Under assumptions 1, 2, 3 and 4, when total iteration $K > \frac{2\kappa_A \theta_A^2}{1 - \beta_A}$, there exists a learning rate $\alpha$ (see Section C.8 in Appendix C) such that*
>
> $$\frac{1}{K} \sum_{k=0}^{K} \mathbb{E}\| \nabla f(\boldsymbol{w}^{(k)}) \|^2 \lesssim \frac{\sigma \sqrt{L\Delta}}{\sqrt{nK}} + \frac{L\Delta(1 + C_A)}{K},$$
>
> *where $\boldsymbol{w}^{(k)} = \pi_A^\top \mathbf{x}^{(k)}$, $C_A$ is a positive constant decided by the mixing matrix $A$. Proof is in Appendix C.*

**Remark 1.** *Theorem 2 establishes the first linear speedup convergence utilizing solely row-stochastic mixing matrices. The term $\frac{C_A L\Delta}{K+1}$ signifies the influence of the network, where $C_A$ is a rational function of $\kappa_A, \beta_A$ and $\theta_A$.*

## 5. Achieving Near-Optimal Convergence Rate

Comparing Theorem 2 with the lower bound in Theorem 1, we identify two key discrepancies preventing PULL-DIAG-GT from achieving the lower bound. First, Theorem 2 relies on Assumption 4, which the lower bound does not require. Second, the constant $C_A$ in Theorem 2 depends on $\theta_A$, the upper bound of the diagonals, which is absent from the lower bound and can grow arbitrarily large even for fixed $\beta_A$ and $\kappa_A$. This section introduces a variant of PULL-DIAG-GT to address these discrepancies and achieve the lower bound.

**Removing $\theta_A$ with multiple gossips.** The requirement for $\theta_A$ (and Assumption 4) arises from the use of $\text{Diag}(A^k)^{-1}$ for gradient correction in the PULL-DIAG-GT update (7b). For small values of $k$, the diagonal elements of $A^k$ can become extremely small due to network sparsity, leading to significant instability in the inversion $\text{Diag}(A^k)^{-1}$ during the initial phase. As $k$ increases, $\text{Diag}(A^k)$ converges to $\text{diag}(\pi_A)$, which stabilizes the correction. This behavior is formally stated in the following lemma:

**Lemma 7.** *For $A \in \mathbb{R}^{n \times n}$ satisfying Assumption 1, if $k \geq \frac{2\ln(\kappa_A) + 2\ln(n)}{1 - \beta_A}$, we have*

$$[A^k]_{ii} > 0 \quad and \quad [A^k]_{ii}^{-1} \leq 2n\kappa_A, \quad \forall i \in [n].$$

Lemma 7 implies that for sufficiently large $k$, the diagonals of $A^k$ are naturally bounded without any extra assumptions. This motivates the use of the multiple gossip strategy to eliminate Assumption 4 and the dependence on $\theta_A$ in Theorem 2. Instead of a single gossip step $\mathbf{z} \xmapsto{\text{A-Protocol}} A\mathbf{z}$, we perform $R$ consecutive gossip steps $\mathbf{z} \xmapsto{\text{Multiple Gossip}} A^R \mathbf{z}$ during each communication phase, where the gossip round $R$ is determined as indicated in Lemma 7.

**Pull-DIAG-GT with multiple gossips.** We now introduce MG-PULL-DIAG-GT to remove Assumption 4 and the the reliance on $\theta_A$. Here, "MG" is short for multiple gossips.

$$\mathbf{x}^{(t+1)} = A^R(\mathbf{x}^{(t)} - \alpha \mathbf{y}^{(t)}) \tag{12a}$$

$$\mathbf{g}^{(t+1)} = \frac{1}{R} \sum_{r=1}^{R} \nabla F(\mathbf{x}^{(t+1)}, \boldsymbol{\xi}^{(t+1,r)}) \tag{12b}$$

$$\mathbf{y}^{(t+1)} = A^R(\mathbf{y}^{(t)} + D_{t+1}^{-1} \mathbf{g}^{(t+1)} - D_t^{-1} \mathbf{g}^{(t)}) \tag{12c}$$

Here $D_t = \text{Diag}(A^{tR}), \forall t \geq 1, D_0 = I_n, \mathbf{y}^{(0)} = \mathbf{g}^{(0)} = \frac{1}{R} \sum_{r=1}^{R} \nabla F(\mathbf{x}^{(0)}, \boldsymbol{\xi}^{(0,r)})$. The implementation details are provided in Appendix B. It is observed that for each iteration $t$, recursions (12a) and (12c) incur $R$ rounds of communication, and (12b) requires $R$ samples to compute the mini-batch stochastic gradient. To ensure a fair comparison with PULL-DIAG-GT, for each $K$-iteration run of PULL-DIAG-GT, we run MG-PULL-DIAG-GT for $T = K/R$ **iterations**, thereby fixing the total number of communication rounds and data samples at $K$.

**Achieving optimal convergence rate.** Technically, by performing multiple gossip steps, we improve the spectral parameter from $\beta_A$ to $\beta_A^R$, which exponentially reduces all terms associated with decentralized communication. Additionally, by utilizing an $R$-mini-batch stochastic gradient, we reduce the gradient variance from $\sigma^2$ to $\sigma^2/R$. However, reducing the outer iterations from $K$ to $K/R$ may polynomially slow the convergence. By carefully balancing this exponential gain against the polynomial cost with an appropriately chosen $R$, we can improve overall convergence, ultimately achieving optimal performance:

**Theorem 3.** *Suppose Assumptions 1,2 and 3 hold, and set $T = K/R$. When $R = \lceil \frac{3(1+\ln(\kappa_A)+\ln(n))}{1-\beta_A} \rceil$ and $\alpha$ being selected properly, we have*

$$\frac{1}{T} \sum_{t=1}^{T} \mathbb{E}[\|\nabla f(\boldsymbol{w}^{(t)})\|_F^2]$$
$$\lesssim \frac{\sigma\sqrt{L\Delta}}{\sqrt{nK}} + \frac{(1+\ln(\kappa_A)+\ln(n))L\Delta}{(1-\beta_A)K},$$

*where $\boldsymbol{w}^{(k)} = \pi_A^\top \mathbf{x}^{(k)}$. The proof is in Appendix D.*

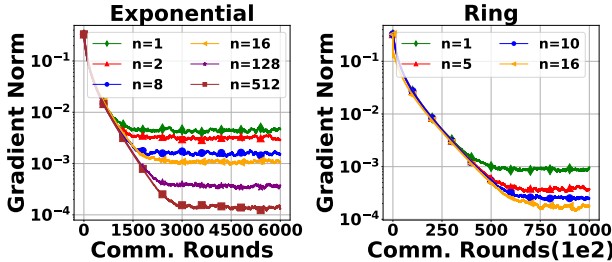

*Figure 2.* Performance of PULL-DIAG-GT for non-convex logistic regression evaluated across exponential graphs and ring graphs. Number $n$ denotes the number of nodes.

**Remark 2.** *It is observed that Theorem 3 is independent of Assumption 4 and $\theta_A$ due to the multiple gossip strategy.*

**Remark 3.** *When $\ln(n)$ is negligible compared to $\ln(\kappa_A)$, Theorem 3 aligns with our lower bound (Theorem 1), making both the lower bound and the algorithm optimal. Otherwise, we say that MG-PULL-DIAG-GT achieves near-optimal complexity (rather than optimal complexity) due to the existence of the logarithmic gap $\ln(n)$.*

## 6. Experiments

In this section, we empirically validate the theoretical results presented in Theorems 2 and 3. For the stochastic gradient oracle, we focus on the case where each node has access to a finite dataset, and the stochastic gradient is computed with respect to a randomly chosen data sample at each iteration. To assess the performance of the algorithms, we conduct experiments on a synthetic dataset, MNIST dataset and CIFAR-10 dataset. Implementation details are provided in Appendix E for reference.

### 6.1. Non-convex Logistic Regression for Classification

In this first group of experiment, we minimize a synthetic nonconvex loss function (Antoniadis et al., 2011; Xin et al., 2021; Alghunaim & Yuan, 2022; Liang et al., 2023) that satisfies the $L$-smooth property. Our experiments are conducted on directed exponential graphs (Xin et al., 2021; Ying et al., 2021) and directed ring graphs (see Figure A1 in Appendix E.1). For exponential graphs, we evaluate the performance across network sizes of 1 (single node), 2, 8, 16, 128 and 512. For ring graphs, we evaluate the performance across network sizes of 1 (single node), 5, 10, 16.

The results in Figure 2 reveal that, for each fixed topology, the gradient curve decreases proportionally to the square root of the number of nodes after the same number of communication rounds. This numerically validates our Theorem 2 that PULL-DIAG-GT is able to achieve linear speedup.

### 6.2. Neural Network for Multi-Class Classification

In the second group of experiment, we focus on a digit-classification task using the MNIST dataset. We evaluate the performance of MG-PULL-DIAG-GT against the vanilla PULL-DIAG-GT across four distinct network topologies: a ring graph, an undirected grid graph, a geometric graph, and a nearest neighbor graph, each comprising 16 nodes. These topologies are illustrated in Figure A1 in Appendix E.1. The weights of the mixing matrices are determined using the Metropolis rule (Nedić et al., 2018), which produces row-stochastic but not doubly-stochastic matrices.

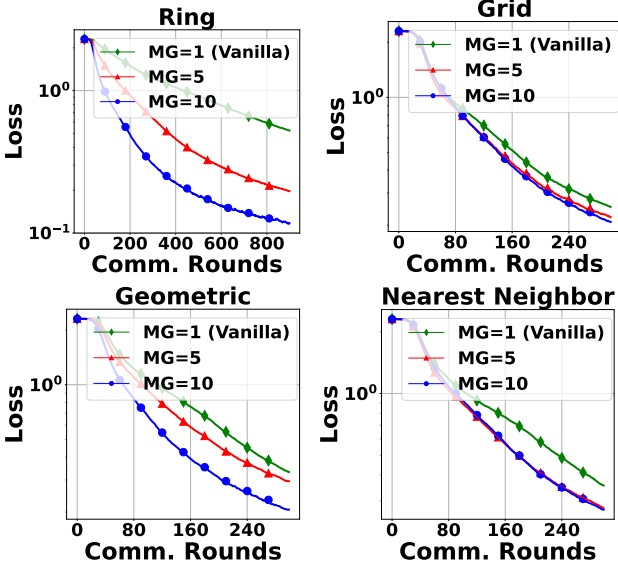

*Figure 3.* Averaged training loss of neural networks on MNIST dataset. Networks trained using MG-PULL-DIAG-GT and vanilla PULL-DIAG-GT. Here, "MG" denotes the number of gossip steps.

Figure 3 demonstrates that MG-PULL-DIAG-GT achieves a consistently faster convergence rate in training loss across all tested topologies, while the corresponding test accuracy is detailed in Figure A2 in Appendix E.

## 6.3. Neural Network for Image Classification

In the third set of experiments, we conducted training of the ResNet-18 model (He et al., 2016) on the CIFAR-10 dataset using a distributed approach. Consistent with our previous MNIST dataset experiment, we evaluated and compared the performance of MG-PULL-DIAG-GT against the standard PULL-DIAG-GT over different topologies. Figure 4

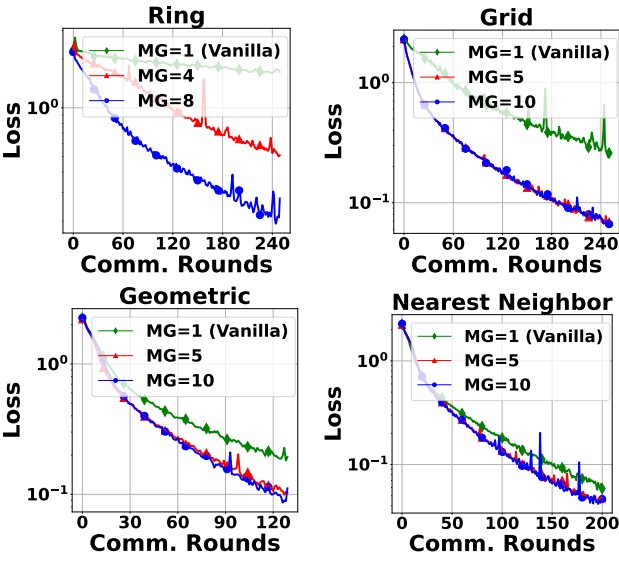

*Figure 4.* Averaged training loss of neural networks on CIFAR-10 dataset. Networks trained using MG-PULL-DIAG-GT and vanilla PULL-DIAG-GT. Here, "MG" denotes the number of gossip steps.

illustrates the stability of MG-PULL-DIAG-GT when applied to a larger real-world dataset. In the context of sparse topologies like the ring and grid graphs, MG-PULL-DIAG-GT effectively reduces the influence of sparse structures, resulting in superior performance compared to the vanilla PULL-DIAG-GT. The corresponding test accuracy is detailed in Figure A3 in Appendix E.

## 7. Conclusions and Limitations

In this paper, we investigate nonconvex, stochastic decentralized optimization over row-stochastic networks. We establish the first lower bound on the convergence rate for this setting. Additionally, we present the first linear speedup convergence rate achieved by PULL-DIAG-GT. To further improve performance, we introduce the multiple gossip technique, leading to the development of MG-PULL-DIAG-GT. This algorithm matches our lower bound up to a logarithmic gap, rendering both the lower bound and the algorithm nearly optimal. Numerical experiments validate our theoretical findings and demonstrate the effectiveness of our approach. A main limitation of our work is that the network impact on PULL-DIAG-GT, such as the explicit influence of $\theta_A$, remains unclear, leaving it for future research.

## Acknowledgment

The work is supported by the National Natural Science Foundation of China under Grants 92370121, 12301392, and W2441021.

## Impact Statement

This paper presents work whose goal is to advance the field of Machine Learning. There are many potential societal consequences of our work, none which we feel must be specifically highlighted here.

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

# Content of Appendix

# A. Lower Bound

## A.1. A Matrix Example

**Proposition 8.** *For any $n \geq 2$, there exists a row-stochastic, primitive matrix $A \in \mathbb{R}^{n \times n}$ satisfying $\beta_A = \frac{\sqrt{2}}{2}$ but $\kappa_A = 2^{n-1}$.*

*Proof.* Proposition 2.5 of Liang et al. (2023) tells us that for any $n \geq 2$, there exists a column-stochastic, primitive matrix $W \in \mathbb{R}^{n \times n}$ satisfying $\beta_W = \frac{\sqrt{2}}{2}$ but $\kappa_W = 2^{n-1}$. Taking $A = B^\top$, their Perron vectors are the same, i.e. $\pi_A = \pi_W$. Therefore, $\kappa_A = \kappa_W$. By the definition of the $\pi$-norm, we know that $\beta_A = \|A - A_\infty\|_{\pi_A} = \|\Pi_A^{1/2}(A - A_\infty)\Pi_A^{-1/2}\|_2 = \|(\Pi_W^{-1/2}(W - W_\infty)\Pi_W^{1/2})^\top\|_2 = \|W\|_{\pi_W} = \beta_W$. $\qquad\square$

## A.2. Proof of Theorem 1

The core idea of the proof is derived from (Liang et al., 2023). The first complexity term, $\Omega(\frac{\sigma\sqrt{L\Delta}}{\sqrt{nK}})$, is standard, and its proof can be found in works such as Lu & De Sa (2021) and Yuan et al. (2022). Therefore, we concentrate on proving the second term, $\Omega((1 + \ln(\kappa_A))L\Delta/K)$.

To proceed, let $[x]_j$ represent the $j$-th coordinate of a vector $x \in \mathbb{R}^d$ for $1 \leq j \leq d$, and define:

$$\text{prog}(x) := \begin{cases} 0 & \text{if } x = 0; \\ \max_{1 \leq j \leq d}\{j : [x]_j \neq 0\} & \text{otherwise.} \end{cases}$$

We also introduce several important lemmas, which have been established in previous research.

**Lemma 9** (Lemma 2 of Arjevani et al. (2019))**.** *Consider the function*

$$h(x) := -\psi(1)\phi([x]_1) + \sum_{j=1}^{d-1}\left(\psi(-[x]_j)\phi(-[x]_{j+1}) - \psi([x]_j)\phi([x]_{j+1})\right)$$

*where for any $z \in \mathbb{R}$,*

$$\psi(z) = \begin{cases} 0 & z \leq 1/2; \\ \exp\left(1 - \frac{1}{(2z-1)^2}\right) & z > 1/2, \end{cases} \quad \text{and} \quad \phi(z) = \sqrt{e}\int_{-\infty}^{z} e^{-\frac{1}{2}t^2}\,\mathrm{d}t.$$

*The function $h(x)$ has the following properties:*

1. *$h$ is zero-chain, i.e., $\text{prog}(\nabla h(x)) \leq \text{prog}(x) + 1$ for all $x \in \mathbb{R}^d$.*

2. *$h(x) - \inf_x h(x) \leq \Delta_0 d$, for all $x \in \mathbb{R}^d$ with $\Delta_0 = 12$.*

3. *$h$ is $L_0$-smooth with $L_0 = 152$.*

4. *$\|\nabla h(x)\|_\infty \leq G_\infty$, for all $x \in \mathbb{R}^d$ with $G_\infty = 23$.*

5. *$\|\nabla h(x)\|_\infty \geq 1$ for any $x \in \mathbb{R}^d$ with $[x]_d = 0$.*

**Lemma 10** (Lemma 4 of (Huang et al., 2022))**.** *Letting functions*

$$h_1(x) := -2\psi(1)\phi([x]_1) + 2\sum_{j \text{ even}, 0 < j < d}\left(\psi(-[x]_j)\phi(-[x]_{j+1}) - \psi([x]_j)\phi([x]_{j+1})\right)$$

*and*

$$h_2(x) := 2\sum_{j \text{ odd}, 0 < j < d}\left(\psi(-[x]_j)\phi(-[x]_{j+1}) - \psi([x]_j)\phi([x]_{j+1})\right),$$

*then $h_1$ and $h_2$ satisfy the following properties:*

1. *$\frac{1}{2}(h_1 + h_2) = h$, where $h$ is defined in Lemma 9.*

2. *$h_1$ and $h_2$ are zero-chain, i.e., $\text{prog}(\nabla h_i(x)) \leq \text{prog}(x) + 1$ for all $x \in \mathbb{R}^d$ and $i = 1, 2$. Furthermore, if $\text{prog}(x)$ is odd, then $\text{prog}(\nabla h_1(x)) \leq \text{prog}(x)$; if $\text{prog}(x)$ is even, then $\text{prog}(\nabla h_2(x)) \leq \text{prog}(x)$.*

3. $h_1$ and $h_2$ are also $L_0$-smooth with $L_0 = 152$.

We are now ready to prove our lower bound. This proceeds in three steps. Without loss of generality, we assume $n$ can be divided by 3.

(Step 1.) We let $f_i = L\lambda^2 h_1(x/\lambda)/L_0, \forall i \in E_1 \triangleq \{j : 1 \le j \le n/3\}$ and $f_i = L\lambda^2 h_2(x/\lambda)/L_0, \forall i \in E_2 \triangleq \{j : 2n/3 \le j \le n\}$, where $h_1$ and $h_2$ are defined in Lemma 10, and $\lambda > 0$ will be specified later. By the definitions of $h_1$ and $h_2$, we have that $f_i, \forall 1 \le i \le n$, is zero-chain and $f(x) = n^{-1} \sum_{i=1}^n f_i(x) = 2L\lambda^2 h(x/\lambda)/3L_0$. Since $h_1$ and $h_2$ are also $L_0$-smooth, $\{f_i\}_{i=1}^n$ are $L$-smooth. Furthermore, since

$$f(0) - \inf_x f(x) = \frac{2L\lambda^2}{3L_0}(h(0) - \inf_x h(x)) \le \frac{L\lambda^2 \Delta_0 d}{L_0},$$

to ensure $\{f_i\}_{i=1}^n$ satisfy L-smooth Assumption, it suffices to let

$$\frac{L\lambda^2 \Delta_0 d}{L_0} \le \Delta, \quad i.e., \quad \lambda \le \sqrt{\frac{L_0 \Delta}{L \Delta_0 d}}. \tag{13}$$

With the functions defined above, we have $f(x) = n^{-1} \sum_{i=1}^n f_i(x) = L\lambda^2 l(x/\lambda)/(3L_0)$ and $\text{prog}(\nabla f_i(x)) = \text{prog}(x) + 1$ if $\text{prog}(x)$ is even and $i \in E_1$ or $\text{prog}(x)$ is odd and $i \in E_2$, otherwise $\text{prog}(\nabla f_i(x)) \le \text{prog}(x)$. Therefore, to make progress (i.e., to increase $\text{prog}(x)$), for any gossip algorithm $\mathbb{A} \in \mathcal{A}_W$, one must take the gossip communication protocol to transmit information between $E_1$ and $E_2$ alternatively.

(Step 2.) We consider the noiseless gradient oracles and the constructed mixing matrix $W$ in Subsection 8 with $\epsilon = 2\beta_A^2 - 1$ so that $\frac{1+\ln(\kappa_A)}{1-\beta_A} = O(n)$. Note the directed distance from $E_1$ to $E_2$ is $n/3$. Consequently, starting from $x^{(0)} = 0$, it takes of at least $n/3$ communications for any possible algorithm $\mathbb{A} \in \mathcal{A}_A$ to increase $\text{prog}(\hat{x})$ by 1 if it is odd. Therefore, we have $\lceil \text{prog}(\hat{x}^{(k)})/2 \rceil \le \lfloor \frac{k}{2n/3} \rfloor, \forall k \ge 0$. This further implies

$$\text{prog}(\hat{x}^{(k)}) \le 2 \left\lfloor \frac{k}{2n/3} \right\rfloor + 1 \le 3k/n + 1, \quad \forall k \ge 0. \tag{14}$$

(Step 3.) We finally show the error $\mathbb{E}[\|\nabla f(x)\|^2]$ is lower bounded by $\Omega\left(\frac{(1+\ln(\kappa_A))L\Delta}{(1-\beta_A)K}\right)$, with any algorithm $\mathbb{A} \in \mathcal{A}_W$ with $K$ communication rounds. For any $K \ge n$, we set $d = 2 \left\lfloor \frac{K}{2n/3} \right\rfloor + 2 \le 3K/n + 2 \le 5K/n$ and $\lambda = \left(\frac{nL_0\Delta}{5L\Delta_0 K}\right)^{1/2}$. Then (13) naturally holds. Since $\text{prog}(\hat{x}^{(K)}) < d$ by (14), using the last point of Lemma 9 and the value of $\lambda$, we obtain

$$\mathbb{E}[\|\nabla f(\hat{x})\|^2] \ge \min_{[\hat{x}]_d=0} \|\nabla f(\hat{x})\|^2 \ge \frac{L^2\lambda^2}{9L_0^2} = \Omega\left(\frac{nL\Delta}{K}\right).$$

Finally, by using $n = \Omega((1 + \ln(\kappa_A))/(1 - \beta_A))$, we complete the proof of Theorem 1.

## B. Algorithm Implementation Details

We provide complete expression and pseudo code for PULL-DIAG protocol, PULL-DIAG-GT and MG-PULL-DIAG-GT here.

For PULL-DIAG protocol, the iteration is:

$$V_{k+1} = AV_k, \quad D_{k+1} = \text{Diag}(nV_{k+1}),$$
$$\mathbf{z}^{(k+1)} = n^{-1}V_{k+1}D_{k+1}^{-1}\mathbf{z}.$$

where $V_0 = I_n$.

For PULL-DIAG-GT, the iteration is:

$$\mathbf{x}^{(k+1)} = A(\mathbf{x}^{(k)} - \alpha\mathbf{y}^{(k)}) \tag{16a}$$
$$\tilde{D}_{k+1} = A\tilde{D}_k \tag{16b}$$
$$D_k = \text{Diag}(\tilde{D}_k) \tag{16c}$$
$$\mathbf{y}^{(k+1)} = A(\mathbf{y}^{(k)} + D_{k+1}^{-1}\mathbf{g}^{(k+1)} - D_k^{-1}\mathbf{g}^{(k)}) \tag{16d}$$

---

**Algorithm 1** PULL-DIAG protocol

---

**Require:** Initialize $v_i^{(0)} = e_i$, $z_i$.
    **for** $k = 0, 1, \ldots, K-1$, each node $i$ in parallel **do**
        Update $v_i^{(k+1)} = \sum_{j \in \mathcal{N}_i^{\mathrm{in}}} a_{ij} v_j^{(k)}$;
    **end for**
    Update $z_i^{(0)} = z_i / n v_{ii}^{(K)}$;
    **for** $k = 0, 1, \ldots, K-1$, each node $i$ in parallel **do**
        Update $z_i^{(k+1)} = \sum_{j \in \mathcal{N}_i^{\mathrm{in}}} a_{ij} z_j^{(k)}$;
    **end for**

---

**Algorithm 2** PULL-DIAG-GT

---

**Require:** Initialize $v_i^{(0,0)} = e_i$, $\mathbf{w}^{(0)} = \mathbf{x}^{(0)}$, $g_i^{(0)} = y_i^{(0)} = \nabla F(x^{(0)}; \xi_i^{(0)})$, the mixing matrix $A = [a_{ij}]_{n \times n}$.
    **for** $k = 0, 1, \ldots, K-1$, each node $i$ in parallel **do**
        Let $\boldsymbol{\phi}^{(k+1)} = \boldsymbol{x}_i^{(k)} - \gamma \boldsymbol{y}_i^{(k)}$;
        Update $\boldsymbol{x}_i^{(k+1)} = \sum_{j \in \mathcal{N}_i^{\mathrm{in}}} a_{ij} \boldsymbol{\phi}_j^{(k+1)}$ and $\boldsymbol{v}_i^{(k+1)} = \sum_{j \in \mathcal{N}_i^{\mathrm{in}}} a_{ij} \boldsymbol{v}_j^{(k)}$;
        Compute $\boldsymbol{g}_i^{(k+1)} = \nabla F(\boldsymbol{x}_i^{(k+1)}; \xi_i^{(k+1)})$;
        Let $\boldsymbol{\psi}_i^{(k+1)} = \boldsymbol{y}_i^{(k)} + [\boldsymbol{v}_i^{(k+1)}]_i^{-1} \boldsymbol{g}_i^{(t+1)} - [\boldsymbol{v}_i^{(k)}]_i^{-1} \boldsymbol{g}_i^{(t)}$;
        Update $\boldsymbol{y}_i^{(k+1)} = \sum_{j \in \mathcal{N}_i^{\mathrm{in}}} a_{ij} \boldsymbol{\psi}_j^{(k+1)}$;
    **end for**

---

where $\tilde{D}_0 = I_n$.

For MG-PULL-DIAG-GT, its iteration runs as:

$$\mathbf{x}^{(t+1)} = A^R(\mathbf{x}^{(t)} - \alpha \mathbf{y}^{(t)}) \tag{17a}$$

$$\tilde{D}_{t+1} = A^R \tilde{D}_t \tag{17b}$$

$$D_t = \mathrm{Diag}(\tilde{D}_t) \tag{17c}$$

$$\mathbf{g}^{(t+1)} = \frac{1}{R} \sum_{r=1}^{R} \nabla F(\mathbf{x}^{(t+1)}, \boldsymbol{\xi}^{(t+1,r)}) \tag{17d}$$

$$\mathbf{y}^{(t+1)} = A^R(\mathbf{y}^{(t)} + D_{t+1}^{-1} \mathbf{g}^{(t+1)} - D_t^{-1} \mathbf{g}^{(t)}) \tag{17e}$$

where $\tilde{D}_0 = I_n$.

## C. Convergence of PULL-DIAG-GT

### C.1. Notation

Most of notations in the proof are the same with notations defined in Section 1. We repeat them as follows:

We denote $\mathbb{1}_n$ as an $n$-dimensional all-ones vector. We define $I_n \in \mathbb{R}^{n \times n}$ as the identity matrix. Throughout the paper, $A$ is always a row-stochastic matrix, *i.e.*, $A\mathbb{1}_n = \mathbb{1}_n$. We denote $[n]$ as the index set $\{1, 2, \ldots, n\}$. We denote $\mathrm{Diag}(A)$ as the the diagonal matrix generated from the diagonal entries of $A$. We denote $\mathrm{diag}(v)$ as the diagonal matrix whose diagonal entries comes from vector $v$. We denote $\pi_A$ as the left Perron vector of $A$. We denote $\Pi_A = \mathrm{diag}(\pi_A)$, $\pi_A$-vector norm $\|v\|_{\pi_A} = \|\Pi_A^{1/2} v\|$ and the induced $\pi_A$-matrix norm as $\|W\|_{\pi_A} = \|\Pi_A^{1/2} W \Pi_A^{-1/2}\|_2$. We define $A_\infty = \mathbb{1}_n \pi_A^\top$, $\beta_A = \|A - A_\infty\|_{\pi_A}$, $\kappa_A = \max(\pi_A)/\min(\pi_A)$. Throughout the paper, we let $\boldsymbol{x}_i^{(k)} \in \mathbb{R}^d$ denote the local model copy at node $i$ at iteration $k$. Furthermore, we define the matrices

---

**Algorithm 3** MG-PULL-DIAG-GT: PULL-DIAG-GT with multi-round gossip

---

**Require:** Initialize $\boldsymbol{v}_i^{(0,0)} = \boldsymbol{e}_i$, $\mathbf{w}^{(0)} = \mathbf{x}^{(0)}$, $\boldsymbol{g}_i^{(0)} = \boldsymbol{y}_i^{(0)} = \frac{1}{R}\sum_{r=1}^{R}\nabla F(x^{(0)};\xi_i^{(0,r)})$, the mixing matrix $A = [a_{ij}]_{n\times n}$, the multi-round number $R$.

  **for** $t = 0, 1, \ldots, (K/R) - 1$, each node $i$ in parallel **do**

    Let $\boldsymbol{\phi}^{(t+1,0)} = \boldsymbol{x}_i^{(t)} - \gamma \boldsymbol{y}_i^{(t)}$;

    **for** $r = 0, 1, \ldots, R - 1$, each node $i$ in parallel **do**

      Update $\boldsymbol{\phi}_i^{(t+1,r+1)} = \sum_{j\in\mathcal{N}_i^{\mathrm{in}}} a_{ij}\boldsymbol{\phi}_j^{(t+1,r)}$ and $\boldsymbol{v}_i^{(t,r+1)} = \sum_{j\in\mathcal{N}_i^{\mathrm{in}}} a_{ij}\boldsymbol{v}_j^{(t,r)}$;

    **end for**

    Update $\boldsymbol{x}_i^{(t+1)} = \boldsymbol{\phi}_i^{(t+1,R)}$, $\boldsymbol{v}_i^{(t+1,0)} = \boldsymbol{v}_i^{(t,R)}$;

    Compute $\boldsymbol{g}_i^{(t+1)} = \frac{1}{R}\sum_{r=1}^{R}\nabla F(bmx_i^{(t+1)};\xi_i^{(t+1,r)})$;

    Let $\boldsymbol{\psi}_i^{(t+1,0)} = \boldsymbol{y}_i^{(t)} + [\boldsymbol{v}_i^{(t+1,0)}]_i^{-1}\boldsymbol{g}_i^{(t+1)} - [\boldsymbol{v}_i^{(t,0)}]_i^{-1}\boldsymbol{g}_i^{(t)}$;

    **for** $r = 0, 1, \ldots, R - 1$, each node $i$ in parallel **do**

      Update $\boldsymbol{\psi}_i^{(t+1,r+1)} = \sum_{j\in\mathcal{N}_i^{\mathrm{in}}} a_{ij}\boldsymbol{\psi}_j^{(t+1,r)}$;

    **end for**

    Update $\boldsymbol{y}_i^{(t+1)} = \boldsymbol{\psi}_i^{(t+1,R)}$;

  **end for**

---

$$\mathbf{x}^{(k)} := [(\boldsymbol{x}_1^{(k)})^\top; (\boldsymbol{x}_2^{(k)})^\top; \cdots; (\boldsymbol{x}_n^{(k)})^\top] \in \mathbb{R}^{n\times d},$$

$$\nabla F(\mathbf{x}^{(k)};\boldsymbol{\xi}^{(k)}) := [\nabla F_1(\boldsymbol{x}_1^{(k)};\xi_1^{(k)})^\top; \cdots; \nabla F_n(\boldsymbol{x}_n^{(k)};\xi_n^{(k)})^\top] \in \mathbb{R}^{n\times d},$$

$$\nabla \mathbf{f}_k := [\nabla f_1(\boldsymbol{x}_1^{(k)})^\top; \nabla f_2(\boldsymbol{x}_2^{(k)})^\top; \cdots; \nabla f_n(\boldsymbol{x}_n^{(k)})^\top] \in \mathbb{R}^{n\times d},$$

by stacking all local variables. The upright bold symbols (e.g. $\mathbf{x}, \mathbf{w}, \mathbf{g} \in \mathbb{R}^{n\times d}$) always denote stacked network-level quantities.

Besides, we define $\overline{\nabla f}(\mathbf{x}^{(k)}) = n^{-1}\mathbb{1}_n^\top \nabla f(\mathbf{x}^{(k)})$, $\Delta_x^{(k)} := (I - A_\infty)\mathbf{x}^{(k)}$, $\Delta_g^{(k)} = \mathbf{g}^{(k+1)} - \mathbf{g}^{(k)}$, $\forall k \geq 0$, $\Delta_g^{(-1)} = \mathbf{g}^{(0)}$.

### C.2. Linear Algebra Inequalities

We outline some useful inequalities which will be frequently used in the proof.

**Lemma 11** (ROLLING SUM LEMMA). *If $l \geq 1$ and $A \in \mathbb{R}^{n\times n}$ is a row-stochastic matrix satisfying Assumption 1, the following estimation holds for $\forall T \geq 0$.*

$$\sum_{k=0}^{T}\|\sum_{i=0}^{k}(A^{k+l-i} - A_\infty)\Delta^{(i)}\|_F^2 \leq s_A^2 \sum_{i=0}^{T}\|\Delta^{(i)}\|_F^2, \tag{18}$$

*where $\Delta^{(i)} \in \mathbb{R}^{n\times d}$ are arbitrary matrices, and $s_A$ is defined by:*

$$s_A := \max_{k\geq 1}\|A^k - A_\infty\|_2 \cdot \frac{1 + \frac{1}{2}\ln(\kappa_A)}{1 - \beta_A}.$$

*Proof.* First, we prove that

$$\|A^i - A_\infty\|_2 \leq \sqrt{\kappa_A}\beta_A^i, \forall i \geq 0. \tag{19}$$

Notice that $\beta_A := \|A - A_\infty\|_{\pi_A}$ and

$$\|A^i - A_\infty\|_{\pi_A} = \|(A - A_\infty)^i\|_{\pi_A} \leq \|A - A_\infty\|_{\pi_A}^i = \beta_A^i,$$

we have

$$\|(A^i - A_\infty)v\| = \|\Pi_A^{-1/2}(A^i - A_\infty)v\|_{\pi_A} \leq \sqrt{\underline{\pi_A}}\beta_A^i\|v\|_{\pi_A} \leq \sqrt{\kappa_A}\beta_A^i\|v\|.$$

The last inequality comes from $\|v\|_{\pi_A} \leq \overline{\pi_A}\|v\|$. Therefore, (19) holds.

Second, we want to prove that for all $k \geq 0$, we have

$$\sum_{i=0}^{k} \|A^{k+l-i} - A_\infty\|_2 \leq s_A. \tag{20}$$

Towards this end, we define $M_A := \max_{k \geq 1} \|A^k - A_\infty\|_2$. According to (19), $M_A$ is well-defined. We also define $p = \max\left\{\frac{\ln(\sqrt{\kappa_A}) - \ln(M_A)}{-\ln(\beta_A)}, 0\right\}$, then we can verify that $\|A^i - A_\infty\|_2 \leq \min\{M_A, M_A\beta_A^{i-p}\}, \forall i \geq 1$. With this inequality, we can bound $\sum_{i=0}^{k} \|A^{k+1-i} - A_\infty\|_2$ as follows:

$$\sum_{i=0}^{k} \|A^{k+1-i} - A_\infty\|_2 = \sum_{i=1}^{\min\{\lfloor p\rfloor, k\}} \|A^i - A_\infty\|_2 + \sum_{i=\min\{\lfloor p\rfloor, k\}+1}^{k+1} \|A^i - A_\infty\|_2$$

$$\leq \sum_{i=0}^{\min\{\lfloor p\rfloor, k\}} M_A + \sum_{i=\min\{\lfloor p\rfloor, k\}+1}^{k+1} M_A\beta_A^{i-p}$$

$$\leq M_A \cdot (1 + \min\{\lfloor p\rfloor, k\}) + M_A \cdot \frac{1}{1-\beta_A}\beta_A^{\min\{\lfloor p\rfloor, k\}+1-p}. \tag{21}$$

If $p = 0$, (21) is simplified to $\sum_{i=0}^{k} \|A^{k+1-i} - A_\infty\|_2 \leq M_A \cdot \frac{1}{1-\beta_A}$ and (20) is naturally satisfied. If $p > 0$, let $x = \min\{\lfloor p\rfloor, k\} + 1 - p \in [0, 1)$, (20) is simplified to

$$\sum_{i=0}^{k} \|A^{k-i} - A_\infty\|_2 \leq M_A(x + p + \frac{\beta_A^x}{1-\beta_A}) \leq M_A(p + \frac{1}{1-\beta_A}).$$

Noting that $p \leq \frac{\frac{1}{2}\ln(\kappa_A)}{1-\beta_A}$, we finish the proof of (20).

Finally, to obtain (18), we use Jensen's inequality. For positive numbers $a_i, i \in [k+1]$ satisfying $\sum_{i=1}^{k+1} a_i = 1$, we have

$$\|\sum_{i=0}^{k}(A^{k+1-i} - A_\infty)\Delta^{(i)}\|_F^2 = \|\sum_{i=0}^{k} a_{k+1-i} \cdot a_{k+1-i}^{-1}(A^{k-i} - A_\infty)\Delta^{(i)}\|_F^2$$

$$\leq \sum_{i=0}^{k} a_{k+1-i}\|a_{k+1-i}^{-1}(A^{k+1-i} - A_\infty)\Delta^{(i)}\|_F^2 \leq \sum_{i=0}^{k} a_{k+1-i}^{-1}\|A^{k+1-i} - A_\infty\|_2^2\|\Delta^{(i)}\|_F^2. \tag{22}$$

By choosing $a_{k+1-i} = (\sum_{i=0}^{k} \|A^{k+1-i} - A_\infty\|_2)^{-1}\|A^{k+1-i} - A_\infty\|_2$ in (22), we obtain that

$$\|\sum_{i=0}^{k}(A^{k+1-i} - A_\infty)\Delta^{(i)}\|_F^2 \leq \sum_{i=0}^{k} \|A^{k+1-i} - A_\infty\|_2 \cdot \sum_{i=0}^{k} \|A^{k+1-i} - A_\infty\|_2\|\Delta^{(i)}\|_F^2. \tag{23}$$

By summing up (23) from $k = 0$ to $T$, we obtain that

$$\sum_{k=0}^{T} \|\sum_{i=0}^{k}(A^{k+1-i} - A_\infty)\Delta^{(i)}\|_F^2 \leq s_A \sum_{k=0}^{T}\sum_{i=0}^{k} \|A^{k+1-i} - A_\infty\|_2\|\Delta^{(i)}\|_F^2$$

$$\leq s_A \sum_{i=0}^{T}(\sum_{k=i}^{T} \|A^{k+1-i} - A_\infty\|_2)\|\Delta^{(i)}\|_F^2 \leq s_A^2 \sum_{i=0}^{T} \|\Delta^{(i)}\|_F^2,$$

which finishes the proof of this lemma. □

**Lemma 12** (CONVERGENCE OF DIAGONAL MATRIX). *The following inequalities hold for all $k \geq 1$.*

1. $\|\pi_A^\top D_k^{-1} - \mathbb{1}_n^\top\| \leq \theta_A\sqrt{n\kappa_A}\beta_A^k$.

2. $\|D_k^{-1} - \Pi_A^{-1}\|_2 \le \theta_A \sqrt{\kappa_A^3 n^3} \beta_A^k$.

3. $\|D_k^{-1} - D_{k+1}^{-1}\|_2 \le 2\theta_A \sqrt{\kappa_A^3 n^3} \beta_A^k$.

*Proof.* Denote $\Pi_A = \text{diag}(A_\infty)$ and $\underline{\pi}_A = \min_i [\pi_A]_i$.

The first conclusion comes from

$$\|\pi_A^\top D_k^{-1} - \mathbb{1}_n^\top\| \le \|D_k^{-1}\| \|\pi_A - \text{diag}(A^k)\| \le \theta_A \|\pi_A - \text{diag}(A^k)\|$$
$$\le \theta_A \|A^k - A_\infty\|_F \le \sqrt{n}\theta_A \|A^k - A_\infty\|_2 \le \theta_A \sqrt{n\kappa_A} \beta_A^k.$$

The second conclusion can be derived from the first conclusion:

$$\|D_k^{-1} - \Pi_A^{-1}\|_2 \le \|\Pi_A^{-1}\|_2 \|\Pi_A D_k^{-1} - I_n\|_2 = \underline{\pi}_A^{-1} \max_i [\pi_A^\top D_k^{-1} - \mathbb{1}_n^\top]_i \le n\kappa_A \|\pi_A^\top D_k^{-1} - \mathbb{1}_n^\top\| \le \theta_A \sqrt{n^3 \kappa_A^3} \beta_A^k.$$

Finally, note that $\|D_k^{-1} - D_{k+1}^{-1}\|_2 \le \|D_k^{-1} - \Pi_A^{-1}\|_2 + \|D_{k+1}^{-1} - \Pi_A^{-1}\|_2$, we obtain the third conclusion. $\qquad\square$

## C.3. Proof of Lemma 2

Left-multiply $\pi_A^\top$ on both sides of (7a), we have $\boldsymbol{w}^{(k+1)} = \boldsymbol{w}^{(k)} - \alpha \pi_A^\top \mathbf{y}^{(k)}$. Using Assumption 2, we know that

$$f(\boldsymbol{w}^{(k+1)}) \le f(\boldsymbol{w}^{(k)}) - n^{-1}\alpha \left\langle n\nabla f(\boldsymbol{w}^{(k)}), \pi_A^\top \mathbf{y}^{(k)} \right\rangle + \frac{\alpha^2 L}{2} \|\pi_A^\top \mathbf{y}^{(k)}\|^2. \tag{24}$$

By taking expectations on $\mathcal{F}_{(k)}$ both sides of (24), we have

$$\mathbb{E}[f(\boldsymbol{w}^{(k+1)})|\mathcal{F}_k] \le f(\boldsymbol{w}^{(k)}) - n^{-1}\alpha \mathbb{E}\left[\left\langle n\nabla f(\boldsymbol{w}^{(k)}), \pi_A^\top \mathbf{y}^{(k)} \right\rangle | \mathcal{F}_k\right] + \frac{\alpha^2 L}{2} \mathbb{E}[\|\pi_A^\top \mathbf{y}^{(k)}\|^2 | \mathcal{F}_k]$$

$$= f(\boldsymbol{w}^{(k)}) - n^{-1}\alpha \left\langle n\nabla f(\boldsymbol{w}^{(k)}), \pi_A^\top D_k^{-1}\nabla f(\mathbf{x}^{(k)}) \right\rangle + \frac{\alpha^2 L}{2} \|\pi_A^\top D_k^{-1}\nabla f(\mathbf{x}^{(k)})\|^2$$

$$\quad + \frac{\alpha^2 L}{2} \text{Var}[\pi_A^\top D_k^{-1}\nabla f(\mathbf{x}^{(k)})]$$

$$\overset{(a)}{\le} f(\boldsymbol{w}^{(k)}) - n^{-1}\alpha \left\langle n\nabla f(\boldsymbol{w}^{(k)}), \pi_A^\top D_k^{-1}\nabla f(\mathbf{x}^{(k)}) \right\rangle$$

$$\quad + \frac{\alpha^2 L}{2} \|\pi_A^\top D_k^{-1}\nabla f(\mathbf{x}^{(k)})\|^2 + \frac{\alpha^2 L\sigma^2}{2} \sum_{j=1}^n \left(\frac{[\pi_A]_j}{[D_k]_j}\right)^2$$

$$= f(\boldsymbol{w}^{(k)}) - \left(\frac{\alpha - n\alpha^2 L}{2n}\right) \mathbb{E}[\|\pi_A^\top D_k^{-1}\nabla f(\mathbf{x}^{(k)})\|^2] - \frac{n\alpha}{2}\|\nabla f(\boldsymbol{w}^{(k)})\|^2$$

$$\quad + \frac{\alpha}{2n} \|n\nabla f(\boldsymbol{w}^{(k)}) - \pi_A^\top D_k^{-1}\nabla f(\mathbf{x}^{(k)})\|^2 + \frac{\alpha^2 L\sigma^2}{2} d_k$$

$$\overset{\alpha \le \frac{1}{2nL}}{\le} f(\boldsymbol{w}^{(k)}) - \frac{\alpha}{4n}\|\pi_A^\top D_k^{-1}\nabla f(\mathbf{x}^{(k)})\|^2 - \frac{n\alpha}{2}\|\nabla f(\boldsymbol{w}^{(k)})\|^2$$

$$\quad + n\alpha \|\nabla f(\boldsymbol{w}^{(k)}) - n^{-1}\mathbb{1}_n^\top \nabla f(\mathbf{x}^{(k)})\|^2 + \frac{\alpha}{n}\|(\mathbb{1}_n^\top - \pi_A^\top D_k^{-1})\nabla f(\mathbf{x}^{(k)})\|^2 + \frac{\alpha^2 L\sigma^2}{2} d_k$$

where (a) holds because the gradient noise is linearly independent, $\text{Var}[\pi_A^\top D_k^{-1}\nabla f(\mathbf{x}^{(k)})] = \text{Var}[\sum_{j=1}^n \frac{[\pi_A]_j}{[D_k]_{jj}}\nabla F_j(\boldsymbol{x}_j;\xi_j)] = \sum_{j=1}^n (\frac{[\pi_A]_j}{[D_k]_{jj}})^2 \text{Var}[\nabla F_j(\boldsymbol{x}_j;\xi_j)] \le \sigma^2 \sum_{j=1}^n (\frac{[\pi_A]_j}{[D_k]_{jj}})^2 = \sigma^2 d_k$. $d_k$ is defined as $d_k = \sum_{j=1}^n (\frac{[\pi_A]_j}{[D_k]_{jj}})^2$.

Note that $\|\nabla f(\boldsymbol{w}^{(k)}) - n^{-1}\mathbb{1}_n^\top \nabla f(\mathbf{x}^{(k)})\|^2 = \frac{1}{n^2}\|\mathbb{1}_n^\top(\nabla f(\boldsymbol{w}^{(k)}) - \nabla f(\mathbf{x}^{(k)}))\|^2 \le \frac{1}{n}\sum_{i=1}^n \|\nabla f_i(\boldsymbol{x}_i^{(k)}) - \nabla f_i(\boldsymbol{w}^{(k)})\|^2 \le n^{-1}L^2 \sum_{i=1}^n \|\boldsymbol{x}_i^{(k)} - \boldsymbol{w}^{(k)}\|^2 = n^{-1}L^2\|\Delta_x^{(k)}\|_F^2$, we thus obtain Lemma 2. $\qquad\square$

## C.4. Estimate Descent Deviation

Using the first statement of Lemma 12, we know that

$$\|\pi_A^\top \text{Diag}(A^k)^{-1} - \mathbb{1}_n^\top\| \leq \theta_A \sqrt{n \kappa_A} \beta_A^k, \quad \forall k \geq 1.$$

Next, with Assumption 2 we know that $\forall \boldsymbol{x}, \boldsymbol{y} \in \mathbb{R}^d, i \in [n]$,

$$f_i(\boldsymbol{y}) \leq f_i(\boldsymbol{x}) + \langle \nabla f_i(\boldsymbol{x}), \boldsymbol{y} - \boldsymbol{x} \rangle + \frac{L}{2} \|\boldsymbol{y} - \boldsymbol{x}\|^2.$$

By taking $\boldsymbol{y} = \boldsymbol{x} - \frac{1}{L} \nabla f_i(\boldsymbol{x})$, we obtain that $\frac{1}{2L} \|\nabla f(\boldsymbol{x})\|^2 \leq f_i(\boldsymbol{x}) - f_i(\boldsymbol{y}) \leq f_i(\boldsymbol{x}) - f_i^*$. Furthermore, using $L$-smoothness property and Cauchy-Schwarz inequality, we have

$$\begin{aligned}
\|\nabla f(\mathbf{x}^{(k)})\|_F^2 &\leq 2\|\nabla f(\mathbf{x}^{(k)}) - \nabla f(\mathbf{w}^{(k)})\|_F^2 + 2\|\nabla f(\mathbf{w}^{(k)})\|_F^2 \\
&\leq 2L^2 \|\Delta_x^{(k)}\|_F^2 + 2\sum_{i=1}^n \|\nabla f_i(\boldsymbol{w}^{(k)})\|^2 \\
&\leq 2L^2 \|\Delta_x^{(k)}\|_F^2 + 4L\sum_{i=1}^n (f_i(\boldsymbol{w}^{(k)}) - f_i^*) \\
&= 2L^2 \|\Delta_x^{(k)}\|_F^2 + 4nL(f(\boldsymbol{w}^{(k)}) - f^*),
\end{aligned}$$

By combining the two parts, we complete the proof of Lemma 3.

## C.5. Absorb extra $f(w) - f^*$

**Lemma 13.** *For any* $\Delta_k, S_k, F_k, c \in \mathbb{R}^+, \beta \in [0,1)$, *if*

$$S_k \leq (1 + c\alpha\beta^k)\Delta_k - \Delta_{k+1} + F_k, \quad \forall k \geq 1,$$

*then, by selecting* $\alpha \leq \frac{1-\beta}{c\beta}$, *we obtain*

$$\sum_{k=1}^K S_k \leq 3\Delta_1 + 3\sum_{k=1}^K F_k.$$

Suppose that $S_k \leq (1 + \alpha c\beta^k)\Delta_k - \Delta_{k+1} + F_k, \forall k \geq 1$, Define $U_k = \prod_{i=1}^k (1 + \alpha c\beta^i), \forall k \geq 1, U_0 = 1$. Then we have

$$U_k^{-1} S_k \leq U_{k-1}^{-1}\Delta_k - U_k^{-1}\Delta_{k+1} + U_k^{-1}F_k, \forall k \geq 1. \tag{25}$$

By summing up (25) from $k = 1$ to $K$, we have

$$\sum_{k=1}^K U_k^{-1} S_k \leq \Delta_1 - U_K^{-1}\Delta_{K+1} + \sum_{k=1}^K U_k^{-1}F_k.$$

When $\alpha < \frac{1-\beta}{c\beta} \triangleq \alpha_2$, we have $U_k = \exp(\sum_{i=1}^k \ln(1 + \alpha c\beta^i)) \leq \exp(\sum_{i=1}^k \alpha c\beta^i) \leq \exp(\frac{\alpha c\beta}{1-\beta}) \leq e < 3$. Therefore,

$$\sum_{k=1}^K S_k \leq U_K(\sum_{k=1}^K U_k^{-1}S_k) \leq U_K(\Delta_1 - U_K^{-1}\Delta_{k+1} + \sum_{k=1}^K U_k^{-1}F_k) \leq 3\Delta_1 + 3\sum_{k=1}^K F_k.$$

$\square$

## C.6. Proof of Lemma 4

The following lemma is the formal version of Lemma 4.

**Lemma 14.** *For any $k \geq 0$, we have*

$$\Delta_x^{(k+1)} = -\alpha \sum_{i=0}^{k} (A - A_\infty)^{k+1-i} \Delta_y^{(i)}.$$

$$\Delta_y^{(k+1)} = \sum_{i=0}^{k} (A - A_\infty)^{k+1-i} D_{i+1}^{-1} \Delta_g^{(i)}$$

$$+ \sum_{i=0}^{k} \sum_{l=i}^{k} (A - A_\infty)^{k+1-l} (D_{l+1}^{-1} - D_l^{-1}) \Delta_g^{(i-1)}.$$

We prove this lemma by induction. Easy to verify that the transformation holds for $k = 0$. Suppose the transformation holds for $k - 1$, then we have

$$(I - A_\infty) \mathbf{y}^{(k+1)} = (A - A_\infty)(I - A_\infty) \mathbf{y}^{(k)} + (A - A_\infty)(D_{k+1}^{-1} \mathbf{g}^{(k+1)} - D_k^{-1} \mathbf{g}^{(k)})$$

$$= \sum_{i=0}^{k-1} (A - A_\infty)^{k+1-i} D_{i+1}^{-1} \Delta_g^{(i)} + \sum_{i=0}^{k-1} \sum_{l=i}^{k-1} (A - A_\infty)^{k+1-l} (D_{l+1}^{-1} - D_l^{-1}) \Delta_g^{(i-1)}$$

$$+ (A - A_\infty) D_{k+1}^{-1} \Delta_g^{(k)} + (A - A_\infty)(D_{k+1}^{-1} - D_k^{-1}) \mathbf{g}^{(k)}$$

$$= \sum_{i=0}^{k} (A - A_\infty)^{k+1-i} D_{i+1}^{-1} \Delta_g^{(i)} + \sum_{i=0}^{k} \sum_{l=i}^{k} (A - A_\infty)^{k+1-l} (D_{l+1}^{-1} - D_l^{-1}) \Delta_g^{(i-1)}$$

which finishes the proof. $\square$

## C.7. Proof of Lemma 6

We start with the following Lemma.

**Lemma 15** (Consensus Lemma for y ).

$$\sum_{k=0}^{K} \mathbb{E}[\|(I - A_\infty) \mathbf{y}^{(k+1)}\|^2]_F \leq C_{y,\sigma}(K+1)\sigma^2 + \alpha^2 C_{y,y} \sigma^2 \sum_{k=0}^{K} d_k + C_{y,0} \|\nabla f(\mathbf{x}^{(0)})\|_F^2$$

$$+ C_{y,x} L^2 \sum_{k=0}^{K} \mathbb{E}\|\Delta_x^{(k+1)}\|_F^2 + \alpha^2 L^2 C_{y,y} \sum_{k=0}^{K} \mathbb{E}\|\pi_A^\top D_k^{-1} \nabla f(\mathbf{x}^{(k)})\|_F^2 \tag{26}$$

*where* $C_{y,\sigma} = 6 n s_A M_A \theta_A^2 + \frac{8n^2 \kappa_A^2 \beta_A^2}{1 - \beta_A^2}$, $C_{y,0} = \frac{8n \kappa_A^2 \beta_A^2}{(1 - \beta_A^2)^3}$, $C_{y,x} = 18 s_A^2 \theta_A^2 + \frac{144 n \kappa_A^2 \beta_A^2}{(1 - \beta_A^2)^4}$, $C_{y,y} = 9 n s_A^2 \theta_A^2 + \frac{72 n^2 \kappa_A^2 \beta_A^2}{(1 - \beta_A^2)^4}$.

*Proof.* Using Lemma 4, we know that $(I - A_\infty) \mathbf{y}^{(k+1)}$ can be decomposed to two rolling sums $\sum_{i=0}^{k} (A - A_\infty)^{k+1-i} D_{i+1}^{-1} \Delta_g^{(i)}$ and $\sum_{i=0}^{k} \sum_{l=i}^{k} (A - A_\infty)^{k+1-l} (D_{l+1}^{-1} - D_l^{-1}) \Delta_g^{(i-1)}$. For the first rolling sum, using Assumption 3 that noise are linearly independent, we can divide $\Delta_g^{(i)}$ into $\nabla f(\mathbf{x}^{(i+1)}) - \nabla f(\mathbf{x}^{(i)})$ and two noise parts, thus applying the Cauchy-Schwarz inequality:

$$\mathbb{E}\| \sum_{i=0}^{k} (A - A_\infty)^{k+1-i} D_{i+1}^{-1} \Delta_g^{(i)} \|_F^2 \leq 6n \mathbb{E} \sum_{i=0}^{k} \|(A - A_\infty)^{k+1-i} D_{i+1}^{-1}\|_2^2 \sigma^2 \tag{27}$$

$$+ 3\mathbb{E}\| \sum_{i=0}^{k} (A - A_\infty)^{k+1-i} D_{i+1}^{-1} (\nabla f(\mathbf{x}^{(i+1)}) - \nabla f(\mathbf{x}^{(i)})) \|_F^2$$

Sum up (27) from $k = 0$ to $K$, we have:

$$\sum_{k=0}^{K} \mathbb{E}\| \sum_{i=0}^{k} (A - A_\infty)^{k+1-i} D_{i+1}^{-1} \Delta_g^{(i)} \|_F^2 \tag{28}$$

$$\leq 6n\sigma^2 \sum_{k=0}^{K} \sum_{i=0}^{k} \mathbb{E}\|(A - A_\infty)^{k+1-i} D_{i+1}^{-1}\|_2^2$$

$$+ 3 \sum_{k=0}^{K} \mathbb{E}\| \sum_{i=0}^{k} (A - A_\infty)^{k+1-i} D_{i+1}^{-1} (\nabla f(\mathbf{x}^{(i+1)}) - \nabla f(\mathbf{x}^{(i)})) \|_F^2$$

$$\leq 6ns_A M_A \theta_A^2 (K+1)\sigma^2 + 3s_A^2 \theta_A^2 \sum_{i=0}^{K} \mathbb{E}\|\nabla f(\mathbf{x}^{(i+1)}) - \nabla f(\mathbf{x}^{(i)})\|_F^2$$

$$\leq 6ns_A M_A \theta_A^2 (K+1)\sigma^2 + 3s_A^2 \theta_A^2 L^2 \sum_{i=0}^{K} \mathbb{E}\|\Delta_x^{(i+1)} - \Delta_x^{(i)} - \alpha A_\infty \mathbf{y}^{(k)}\|_F^2$$

$$\leq 6ns_A M_A \theta_A^2 (K+1)\sigma^2 + 18s_A^2 \theta_A^2 L^2 \sum_{i=0}^{K} \mathbb{E}\|\Delta_x^{(i+1)}\|_F^2 + 9\alpha^2 L^2 s_A^2 \theta_A^2 \sum_{i=0}^{K} \mathbb{E}\|A_\infty \mathbf{y}^{(i)}\|_F^2,$$

where the second inequality uses the rolling sum lemma, the third inequality uses the L-smooth assumption, the final inequality uses Cauchy-Schwartz inequality.

Next, we consider the second part. Similar to (27), we have

$$\mathbb{E}\| \sum_{i=0}^{k} \sum_{l=i}^{k} (A - A_\infty)^{k+1-l} (D_{l+1}^{-1} - D_l^{-1}) \Delta_g^{(i-1)} \|_F^2 \tag{29}$$

$$\leq 8n\sigma^2 \sum_{i=0}^{k} \| \sum_{l=i}^{k} (A - A_\infty)^{k+1-l} (D_{l+1}^{-1} - D_l^{-1}) \|_2^2$$

$$+ 4\mathbb{E}\| \sum_{i=1}^{k} \sum_{l=i}^{k} (A - A_\infty)^{k+1-l} (D_{l+1}^{-1} - D_l^{-1}) (\nabla f(\mathbf{x}^{(i)}) - \nabla f(\mathbf{x}^{(i-1)})) \|_F^2$$

$$+ 4\mathbb{E}\| \sum_{l=0}^{k} (A - A_\infty)^{k+1-l} (D_{l+1}^{-1} - D_l^{-1}) \nabla f(\mathbf{x}^{(0)}) \|_F^2$$

$$\leq 8n\sigma^2 \sum_{i=0}^{k} n\kappa_A^2 \beta_A^{2k+2} + 4k \sum_{i=1}^{k} \| \sum_{l=i}^{k} (A - A_\infty)^{k+1-l} (D_{l+1}^{-1} - D_l^{-1}) \|_2^2 \mathbb{E}\|(\nabla f(\mathbf{x}^{(i)}) - \nabla f(\mathbf{x}^{(i-1)})) \|_F^2$$

$$+ 4\| \sum_{l=0}^{k} (A - A_\infty)^{k+1-l} (D_{l+1}^{-1} - D_l^{-1}) \|_2^2 \|\nabla f(\mathbf{x}^{(0)})\|_F^2$$

$$\leq 8n^2\sigma^2 (k+1)\kappa_A^2 \beta_A^{2k+2} + 4k^3 \sum_{i=1}^{k} n\kappa_A^2 \beta_A^{2k+2} \mathbb{E}\|\nabla f(\mathbf{x}^{(i)}) - \nabla f(\mathbf{x}^{(i-1)})\|_F^2$$

$$+ 4(k+1)^2 n\kappa_A^2 \beta_A^{2k+2} \|\nabla f(\mathbf{x}^{(0)})\|_F^2,$$

where the first inequality is similar to the first inequality of (27), the second and third inequality use the fact that $\|(A - A_\infty)^{k+1-l} (D_{l+1}^{-1} - D_l^{-1})\|_2^2 \leq n\kappa_A^2 \beta_A^{2k+2}$.

Sum up (29) from $k = 0$ to $K$, we have:

$$\sum_{k=0}^{K} \mathbb{E}\| \sum_{i=0}^{k} \sum_{l=i}^{k} (A - A_\infty)^{k+1-l} (D_{l+1}^{-1} - D_l^{-1}) \Delta_g^{(i-1)} \|_F^2 \tag{30}$$

$$\leq 8n^2 \kappa_A^2 \sigma^2 \sum_{k=0}^{K} (k+1) \beta_A^{2k+2} + 4n \kappa_A^2 \beta_A^2 \|\nabla f(\mathbf{x}^{(0)})\|_F^2 \sum_{k=0}^{K} (k+1)^2 \beta_A^{2k+2}$$

$$+ 4n \kappa_A^2 \beta_A^2 \sum_{k=0}^{K} \sum_{i=1}^{k} k^3 \beta_A^{2k} \mathbb{E}\|\nabla f(\mathbf{x}^{(i)}) - \nabla f(\mathbf{x}^{(i-1)})\|_F^2$$

$$\leq \frac{8n^2 \kappa_A^2 \beta_A^2 (K+1) \sigma^2}{1 - \beta_A^2} + \frac{8n \kappa_A^2 \beta_A^2 \|\nabla f(\mathbf{x}^{(0)})\|_F^2}{(1 - \beta_A^2)^3}$$

$$+ \frac{24n \kappa_A^2 \beta_A^2}{(1 - \beta_A^2)^4} \sum_{i=1}^{K} \mathbb{E}\|\nabla f(\mathbf{x}^{(i)}) - \nabla f(\mathbf{x}^{(i-1)})\|_F^2$$

$$\leq \frac{8n^2 \kappa_A^2 \beta_A^2 (K+1) \sigma^2}{1 - \beta_A^2} + \frac{8n \kappa_A^2 \beta_A^2 \|\nabla f(\mathbf{x}^{(0)})\|_F^2}{(1 - \beta_A^2)^3}$$

$$+ \frac{144n \kappa_A^2 \beta_A^2 L^2}{(1 - \beta_A^2)^4} \sum_{i=1}^{K} \mathbb{E}\|\Delta_x^{(i)}\|_F^2 + \frac{72\alpha^2 L^2 n \kappa_A^2 \beta_A^2}{(1 - \beta_A^2)^4} \sum_{i=1}^{K} \mathbb{E}\|A_\infty \mathbf{y}^{(i)}\|_F^2$$

Finally, from the proof of Lemma 2 we know that $\mathbb{E}[\|A_\infty \mathbf{y}^{(k)}\|_F^2] = n\sigma^2 d_k + n\|\pi_A^\top D_k^{-1} \nabla f(\mathbf{x}^{(k)})\|^2$. By replacing $\mathbb{E}[\|A_\infty \mathbf{y}^{(k)}\|_F^2]$ we obtain the lemma for $\Delta_y$. $\qquad\square$

The following Lemma is a formal version of Lemma 6.

**Lemma 16** (Consensus Lemma for x). *When* $\alpha \leq \frac{1}{s_A L \sqrt{2 C_{y,x}}}$ *and* $K > \frac{2\kappa_A \theta_A^2}{1 - \beta_A}$, *we have*

$$\sum_{k=0}^{K} \mathbb{E}\|\Delta_x^{(k+1)}\|_F^2 \leq 2\alpha^2 s_A^2 (C_{y,\sigma} + 3n\alpha^2 L^2 C_{y,y})(K+1)\sigma^2 + 4n\alpha^2 L s_A^2 C_{y,0} \Delta \tag{31}$$

$$+ 2\alpha^4 s_A^2 L^2 C_{y,y} \sum_{k=0}^{K} \mathbb{E}\|\pi_A^\top D_k^{-1} \nabla f(\mathbf{x}^{(k)})\|_F^2$$

*where* $C_{y,x}$, $C_{y,\sigma}$, $C_{y,0}$ *and* $C_{y,y}$ *are defined as in Lemma 15.*

*Proof.* Note that $\Delta_x^{(k+1)} = -\alpha \sum_{i=0}^{k} (A - A_\infty)^{k+1-i} \Delta_y^{(i)}$, we apply the rolling sum lemma and obtain that

$$\sum_{k=0}^{K} \mathbb{E}\|\Delta_x^{(k+1)}\|_F^2 \leq \alpha^2 s_A^2 \sum_{k=0}^{K} \|\Delta_y^{(k)}\|_F^2 \tag{32}$$

$$\overset{\text{Lemma 15}}{\leq} \alpha^2 s_A^2 C_{y,\sigma}(K+1)\sigma^2 + \alpha^2 s_A^2 C_{y,0} \|\nabla f(\mathbf{x}^{(0)})\|_F^2 + \alpha^4 s_A^2 L^2 \sigma^2 C_{y,y} \sum_{k=0}^{K} d_k$$

$$+ \alpha^2 L^2 s_A^2 C_{y,x} \sum_{k=0}^{K} \mathbb{E}\|\Delta_x^{(k+1)}\|_F^2 + \alpha^4 L^2 s_A^2 C_{y,y} \sum_{k=0}^{K} \mathbb{E}\|A_\infty \mathbf{y}^{(k)}\|_F^2$$

To estimate $\sum_{k=0}^{K} d_k$, notice that

$$d_k = \sum_{j=1}^{n} (\frac{[\pi_A]_j}{[D_k]_j})^2 = \sum_{j=1}^{n} (1 + \frac{2([\pi_A]_j - [D_k]_j)}{[D_k]_j} + \frac{([\pi_A]_j - [D_k]_j)^2}{[D_k]_j^2})$$

$$\leq 2n + 2\|\pi_A^\top D_k^{-1} - \mathbb{1}_n^\top\|^2 \leq 2n + 2\theta_A^2 \kappa_A n \beta_A^{2k}$$

Therefore, $\sum_{k=0}^{K} d_k \leq 2n(K+1) + \frac{2\theta_A^2 \kappa_A n}{1-\beta_A^2}$. When $K \geq \frac{2\kappa_A \theta_A^2}{1-\beta_A}$, we have $\sum_{k=0}^{K} d_k \leq 3n(K+1)$. When $\alpha \leq \frac{1}{s_A L \sqrt{2C_{y,x}}} \triangleq \alpha_3$, $\alpha^2 s_A^2 C_{y,x} \leq 1/2$. Therefore, we can subtract $\frac{1}{2} \sum_{k=0}^{T} \mathbb{E}\|\Delta_x^{(k+1)}\|_F^2$ from both sides of (32). Finally, note that $\|\nabla f(\mathbf{x}^{(0)})\|_F^2 \leq 2nL\Delta$, we obtain the lemma.

$\square$

## C.8. Proof of Theorem 2

Using Lemma 3 to estimate the descent deviation in Lemma 2, we have

$$\frac{n\alpha}{2} \mathbb{E}[\|\nabla f(\boldsymbol{w}^{(k)})\|^2] \leq \mathbb{E}[f(\boldsymbol{w}^{(k)}) - f(\boldsymbol{w}^{(k+1)})] + \alpha L^2(1 + 2\theta_A^2 \kappa_A \beta_A^{2k})\mathbb{E}[\|\Delta_x^{(k)}\|_F^2] + 4\alpha\theta^2 \kappa_A n L \mathbb{E}[(f(\boldsymbol{w}^{(k)}) - f^*))]$$

$$+ \frac{\alpha^2 L\sigma^2}{2} d_k - \frac{\alpha}{4n} \mathbb{E}[\|\pi_A^\top D_k^{-1} \nabla f(\mathbf{x}^{(k)})\|^2], \quad \forall k \geq 1.$$

Then, we can apply Lemma 13, where we set $\beta = \beta_A, c = 4n\theta_A^2 \kappa_A L, \Delta_k = \mathbb{E}[f(\boldsymbol{w}^{(k)}) - f^*], S_k = \frac{n\alpha}{2}\mathbb{E}[\|\nabla f(\boldsymbol{w}^{(k)})\|^2] + \frac{\alpha}{4n}\mathbb{E}[\|\pi_A^\top D_k^{-1} \nabla f(\mathbf{x}^{(k)})\|^2], F_k = \frac{\alpha^2 L\sigma^2}{2} d_k + \alpha L^2(1 + 2\theta_A^2 \kappa_A \beta_A^{2k})\mathbb{E}[\|\Delta_x^{(k)}\|^2]$. When $\alpha \leq \frac{1-\beta_A}{4n\theta_A^2 \kappa_A L}$, we obtain that

$$\frac{n\alpha}{2} \sum_{k=1}^{K} \mathbb{E}[\|\nabla f(\boldsymbol{w}^{(k)})\|^2] \leq 3\mathbb{E}[f(\boldsymbol{w}^{(1)}) - f^*] - \frac{\alpha}{4n} \sum_{k=1}^{K} \mathbb{E}[\|\pi_A^\top D_k^{-1} \nabla f(\mathbf{x}^{(k)})\|^2] \tag{33}$$

$$+ 3\alpha L^2(1 + 2\theta_A^2 \kappa_A \beta_A^2) \sum_{k=1}^{K} \|\Delta_x^{(k)}\|_F^2 + \frac{3\alpha^2 L\sigma^2}{2} \sum_{k=1}^{K} d_k$$

For $k = 0$, note that $\Delta_x^{(0)} = 0$ and $D_0 = I_n$, the descent lemma provides

$$\mathbb{E}[f(\boldsymbol{w}^{(1)}) - f^*] \leq -\frac{n\alpha}{2}\|\nabla f(\boldsymbol{x}^{(0)})\|^2 + f(\boldsymbol{x}^{(0)}) - f^* - \frac{\alpha}{4n}\mathbb{E}[\|\pi_A^\top \nabla f(\mathbf{x}^{(0)})\|^2] + \frac{\alpha}{n}\mathbb{E}[\|(\pi_A^\top - \mathbb{1}_n^\top)\nabla f(\mathbf{x})^{(0)}\|^2] + \frac{\alpha L\sigma^2}{2} d_0$$

$$\leq \Delta - \frac{n\alpha}{2}\|\nabla f(\boldsymbol{x}^{(0)})\|^2 - \frac{\alpha}{4n}\mathbb{E}[\|\pi_A^\top \nabla f(\mathbf{x}^{(0)})\|^2] + 2\alpha L\Delta n^{-1}\|\pi_A^\top - \mathbb{1}_n^\top\|^2 + \frac{\alpha L\sigma^2}{2} d_0$$

$$\leq \Delta - \frac{n\alpha}{2}\|\nabla f(\boldsymbol{x}^{(0)})\|^2 - \frac{\alpha}{4n}\mathbb{E}[\|\pi_A^\top \nabla f(\mathbf{x}^{(0)})\|^2] + 2\alpha L\Delta + \frac{\alpha L\sigma^2}{2} d_0 \tag{34}$$

Using (34) and estimate of $\sum_{k=0}^{K} d_k$ in (33), we obtain that

$$\frac{n\alpha}{2} \sum_{k=0}^{K} \mathbb{E}[\|\nabla f(\boldsymbol{w}^{(k)})\|^2] \leq 3(1 + 2\alpha L)\Delta - \frac{\alpha}{4n} \sum_{k=0}^{K} \mathbb{E}[\|\pi_A^\top D_k^{-1} \nabla f(\mathbf{x}^{(k)})\|^2] \tag{35}$$

$$+ 3\alpha L^2(1 + 2\theta_A^2 \kappa_A \beta_A^2) \sum_{k=1}^{K} \|\Delta_x^{(k)}\|_F^2 + \frac{9n\alpha^2 L\sigma^2}{2}(K+1)$$

Note that $\alpha \leq \frac{1}{2nL}, 3(1 + 2\alpha L) \leq 6$. We further plug in Lemma 16 in (33) and obtain that

$$\frac{1}{K+1} \sum_{k=0}^{K} \mathbb{E}[\|\nabla f(\boldsymbol{w}^{(k)})\|^2]$$

$$\leq \frac{12\Delta}{n\alpha(K+1)} + 15\alpha L\sigma^2 + (12n^{-1}\alpha^4 s_A^2 L^2(1 + \theta_A^2 \kappa_A \beta_A^2)C_{y,y} - 0.5n^{-2})\frac{1}{K+1} \sum_{k=0}^{K} \mathbb{E}[\|\pi_A^\top D_k^{-1} \nabla f(\mathbf{x}^{(k)})\|^2]$$

$$+ 6n^2 s_A^2 \alpha^2 L^2(1 + \theta_A^2 \kappa_A \beta_A^2)(2C_{y,\sigma} + 6n\alpha^2 L^2 C_{y,y})\sigma^2 + \frac{12L^3(1 + 2\theta_A^2 \kappa_A \beta_A^2)\alpha^2 s_A^2 C_{y,0}\Delta}{K+1}. \tag{36}$$

When $\alpha \leq \left(\frac{1}{24ns_A^2 L^4(1+\theta_A^2 \kappa_A \beta_A^2)C_{y,y}}\right)^{\frac{1}{4}} \triangleq \alpha_4$, the coefficient of $\sum_{k=0}^{K} \mathbb{E}[\|\pi_A^\top D_k^{-1} \nabla f(\mathbf{x}^{(k)})\|^2]$ is negative. When $\alpha \leq \min\{\frac{1}{\sqrt{24C_{y,\sigma}(1+\theta_A^2 \kappa_A \beta_A^2)s_A nL}} \triangleq \alpha_5, \left(\frac{1}{72n^3 s_A^2 L^3 C_{y,y}}\right)^{\frac{1}{3}} \triangleq \alpha_6\}$, the coefficient $6n^2 s_A^2 \alpha^2 L^2(1 + \theta_A^2 \kappa_A \beta_A^2)(2C_{y,\sigma} +$

$6n\alpha^2 L^2 C_{y,y}) \le \alpha L$. When $\alpha \le \frac{1}{s_A L\sqrt{(1+2\theta_A^2\kappa_A^2)\beta_A C_{y,0}}} \triangleq \alpha_7$, the last term $\frac{12L^3(1+2\theta_A^2\kappa_A\beta_A^2)\alpha^2 s_A^2 C_{y,0}\Delta}{K+1} \le \frac{L\Delta}{K+1}$. Therefore, with sufficiently small $\alpha$, we have

$$\frac{1}{K+1}\sum_{k=0}^{K}\mathbb{E}[\|\nabla f(w^{(k)})\|^2] \le \frac{12\Delta}{n\alpha(K+1)} + 16\alpha L\sigma^2 + \frac{L\Delta}{K+1}.$$

The $\alpha$ here should be smaller than $\min_{1\le i\le 7}\{\alpha_i\}$, where $\alpha_1 = \frac{1}{2nL}$, $\alpha_2 = \frac{1-\beta_A}{4n\theta_A^2\kappa_A\beta_A L}$, $\alpha_3 = \frac{1}{s_A L\sqrt{2C_{y,x}}}$, $\alpha_4 = \left(\frac{1}{24ns_A^2 L^4(1+\theta_A^2\kappa_A\beta_A^2)C_{y,y}}\right)^{\frac{1}{4}}$, $\alpha_5 = \frac{1}{\sqrt{24C_{y,\sigma}(1+\theta_A^2\kappa_A\beta_A^2)s_A nL}}$, $\alpha_6 = \left(\frac{1}{72n^3 s_A^2 L^3 C_{y,y}}\right)^{\frac{1}{3}}$, $\alpha_7 = \frac{1}{s_A L\sqrt{(1+2\theta_A^2\kappa_A\beta_A^2)C_{y,0}}}$. $C_{y,x}, C_{y,y}, C_{y,\sigma}, C_{y,0}$ are positive constants defined in Lemma 15 and are only decided by the mixing matrix $A$.

Finally, by selecting $\alpha = 1/(\sqrt{\frac{4L\sigma^2 n(K+1)}{3\Delta}} + \sum_{i=1}^{7}\alpha_i^{-1})$ which is smaller than $\min_{1\le i\le 7}\{\alpha_i\}$, we have

$$\frac{1}{K+1}\sum_{k=0}^{K}\mathbb{E}[\|\nabla f(w^{(k)})\|^2] \lesssim \sqrt{\frac{L\Delta\sigma^2}{n(K+1)}} + \frac{\Delta(L + n^{-1}\sum_{i=1}^{7}\alpha_i^{-1})}{K+1}.$$

Define $n^{-1}\sum_{i=1}^{7}\alpha_i^{-1} = LC_A$, where $C_A$ is a positive constant decided by the mixing matrix $A$, we have

$$\frac{1}{K+1}\sum_{k=0}^{K}\mathbb{E}[\|\nabla f(w^{(k)})\|^2] \lesssim \sqrt{\frac{L\Delta\sigma^2}{nK}} + \frac{\Delta L(1+C_A)}{K}.$$

$\qquad\square$

## D. Convergence of MG-PULL-DIAG-GT

Compared to vanilla PULL-DIAG-GT, MG-PULL-DIAG-GT introduces two notable changes: multiple gossip and the utilization of batch-average gradients. By calculating the gradient across $R$ batches, under Assumption 3, it follows that the variance of $g$ decreases from $\sigma^2$ to $\hat{\sigma}^2 = \frac{\sigma^2}{R}$. The term multiple gossip signifies the substitution of $A$ with $A^R$; in this scenario, $\beta_A$, $s_A$, $\theta_A$ and $C_A$ are all modified according to the respective measure of $A^R$. Therefore, we define $\hat{\beta}_A = \beta_{A^R} = \|A^R - A_\infty\|_{\pi_A}$, $\hat{\theta}_A = \sup_{k\ge 1}\{[\text{diag}(A^k R)]_i^{-1}\}$, $\hat{s}_A = s_{A^R} = \sup_{k\ge 1}\|A^k R - A_\infty\|_2 \cdot \frac{2(1+\ln(\kappa_A))}{1-\hat{\beta}_A}$. $\hat{C}_A = C_{A^R}$. We define $\hat{\alpha}_i$ has the same expression with $\alpha_i$, but their $\beta_A$, $s_A$, $\theta_A$ are replaced with $\hat{\beta}_A$, $\hat{s}_A$, $\hat{\theta}_A$, respectively.

Using the conclusion of Theorem 2, we obtain

$$\frac{1}{T+1}\sum_{t=0}^{T}\mathbb{E}[\|\nabla f(\boldsymbol{w}^{(k)})\|^2] \lesssim \frac{\hat{\sigma}\sqrt{L\Delta}}{\sqrt{nT}} + \frac{L\Delta(1+\hat{C}_A)}{T} \stackrel{T=\frac{K}{R}}{=} \frac{\sigma\sqrt{L\Delta}}{\sqrt{nK}} + \frac{L\Delta(1+\hat{C}_A)R}{K} \qquad (37)$$

Next, we demonstrate that when $R = \frac{r}{1-\beta_A}$ and $r > 3 + 3\ln(\kappa_A) + 3\ln(n)$, $\hat{C}_A$ is necessarily an absolute constant. Under our definition, $\hat{C}_A = \sum_{i=1}^{7}(nL\hat{\alpha}_i)^{-1}$, so it suffices to establish that $(nL\hat{\alpha}_i)^{-1}$ is an absolute constant, $\forall 1\le i\le 7$.

For $\hat{\alpha}_1 = \frac{1}{2nL}$, $(nL\hat{\alpha}_1)^{-1} = 2$. Before estimating $(nL\hat{\alpha}_i)^{-1}, \forall 2\le i\le 7$, we show that $1-\hat{\beta}_A = \Omega(1)$, $\hat{\theta}_A \le 2n\kappa_A$.

$$\hat{\beta}_A = \|A^R - A_\infty\|_{\pi_A} \le \|A - A_\infty\|_{\pi_A}^R = \beta_A^R = \exp(r\frac{\ln(\beta_A)}{1-\beta_A}) \le \exp(-r) \le \frac{1}{e^3 n^3\kappa_A^3}, \quad 1-\hat{\beta}_A \in (1-1/e, 1].$$

When $k \ge 1$,

$$[A^{kR}]_{ii} = [\pi_A]_i + [A^{kR} - A_\infty]_{ii} \ge \underline{\pi_A} - \|A^{kR} - A_\infty\|_F \ge \underline{\pi_A} - \sqrt{n\kappa_A}\beta_A^R \ge \frac{1}{n\kappa_A} - \sqrt{n\kappa_A}\exp(-r) \ge \frac{1}{n\kappa_A} - \frac{1}{2n\kappa_A} = \frac{1}{2n\kappa_A}$$

Therefore, $\hat{\theta}_A = \sup_{k\ge 1}\max_i[A^{kR}]_{ii}^{-1} \le 2n\kappa_A$. Additionally, we have

$$\hat{s}_A = \sup_{k \geq 1} \|A^{kR} - A_\infty\|_2 \cdot \frac{2(1 + \ln(\kappa_A))}{1 - \hat{\beta}_A} \leq 4\sqrt{\kappa_A}\beta_A^R(1 + \ln(\kappa_A)) \leq 4\sqrt{\kappa_A}(1 + \ln(\kappa_A))\exp(-r) \leq \frac{1}{n\kappa_A}.$$

Note that $\hat{\alpha}_2 = \frac{1 - \hat{\beta}_A}{4n\hat{\theta}_A^2\kappa_A\hat{\beta}_A L} \geq \frac{\exp(r)}{8n^3\kappa_A^3 L} \geq \frac{1}{nL}$, thus, $n^{-1}L^{-1}\hat{\alpha}_2^{-1} \leq 1$.

For $\hat{\alpha}_3$, we have

$$(nL\hat{\alpha}_3)^{-1} = \frac{\hat{s}_A\sqrt{36\hat{s}_A^2\hat{\theta}_A^2 + \frac{288n\kappa_A^2\hat{\beta}_A^2}{(1-\hat{\beta}_A^2)^4}}}{n} \leq 100\frac{\hat{s}_A^2\kappa_A + \sqrt{n}\kappa_A\hat{\beta}_A}{n} \leq 100.$$

For $\hat{\alpha}_4$, we have

$$(nL\hat{\alpha}_4)^{-1} = \left(\frac{24\hat{s}_A^2(1 + \hat{\theta}_A^2\kappa_A\hat{\beta}_A^2)(9n\hat{s}_A^2\hat{\theta}_A^2 + \frac{72n^2\kappa_A^2\hat{\beta}_A^2}{(1-\hat{\beta}_A^2)^4})}{n^3}\right)^{\frac{1}{4}} \leq 10\left(\frac{\hat{s}_A^4 n^3\kappa_A^2 + n^2\kappa_A^2\exp(-2r)}{n^3}\right)^{\frac{1}{4}} \leq 10.$$

For $\hat{\alpha}_5$, we have

$$(nL\hat{\alpha}_5)^{-1} = \hat{s}_A\sqrt{24(6n\hat{s}_A\hat{M}_A\hat{\theta}_A^2 + \frac{8n^2\kappa_A^2\hat{\beta}_A^2}{1-\hat{\beta}_A^2})(1 + \hat{\theta}_A^2\kappa_A\hat{\beta}_A^2)} \leq 40\hat{s}_A\sqrt{n\hat{s}_A\hat{M}_A\kappa_A^2 + n^2\kappa_A^2\hat{\beta}_A^2} \leq 80.$$

Here we use the fact that $\hat{M}_A = \sup_{k \geq 1}\|A^{kR} - A_\infty\|_2 \leq \sqrt{\kappa_A}\beta_A^R \leq \frac{1}{n\kappa_A}$.

For $\hat{\alpha}_6$, we have

$$(nL\hat{\alpha}_6)^{-1} = \left(72\hat{s}_A^2(9n\hat{s}_A^2\hat{\theta}_A^2 + \frac{72n^2\hat{\kappa}_A^2\hat{\beta}_A^2}{(1-\hat{\beta}_A^2)^4})\right)^{\frac{1}{3}} \leq 36\left(n\hat{s}_A^4\kappa_A^2 + n^2\kappa_A^2\hat{\beta}_A^2\right)^{\frac{1}{3}} \leq 72.$$

For $\hat{\alpha}_7$, we have

$$(nL\hat{\alpha}_6)^{-1} = \hat{s}_A\sqrt{(1 + 2\hat{\theta}_A^2\kappa_A\hat{\beta}_A^2)\frac{8\kappa_A^2\hat{\beta}_A^2}{n(1-\hat{\beta}_A^2)^3}} \leq 14\hat{s}_A\kappa_A\hat{\beta}_A \leq 14.$$

By combining these inequalities, we demonstrate that $\hat{C}_A \leq 300$ is an absolute constant. Finally, notice that $r$ is any constant larger than $1 + 3\ln(\kappa_A) + 3\ln(n)$, therefore, by choosing $R = \lceil\frac{1+3\ln(\kappa_A)+3\ln(n)}{1-\beta_A}\rceil$ in (37), we have

$$\frac{1}{T+1}\sum_{t=0}^{T}\mathbb{E}[\|\nabla f(\boldsymbol{w}^{(k)})\|^2] \lesssim \frac{\sigma\sqrt{L\Delta}}{\sqrt{nK}} + \frac{L\Delta(1+\hat{C}_A)R}{K} \lesssim \frac{\sigma\sqrt{L\Delta}}{\sqrt{nK}} + \frac{L\Delta(1+\ln(\kappa_A)+\ln(n))}{(1-\beta_A)K}$$

Finally, we discuss the range of $T$ which ensures $\sum_{t=0}^{T}d_{tR} \leq 3n(T+1)$. Note that $d_0 = \|\pi_A\|^2 \leq 1$, and for any $t \geq 1$ we have

$$d_{tR} = \|\pi_A^\top D_{tR}^{-1}\|^2 \leq 2\|\mathbb{1}_n\|^2 + 2\|\pi_A^\top D_{tR}^{-1} - \mathbb{1}_n^\top\|^2 \leq 2n + 2\hat{\theta}_A^2\|\pi_A - \text{diag}(A^t R)\|^2$$
$$\leq 2n + 16n^2\kappa_A^2 \cdot n\kappa_A\hat{\beta}_A \leq 2n + 16e^{-3} \leq 3n.$$

So we can cancel the requirement on the range of $T$ or $K$.

$\square$

# E. Experiment Details

In this section, we provide unexplained details of the experimental setup in the article.

## E.1. Network Design

In this subsection, we present the networks used in our experiments.

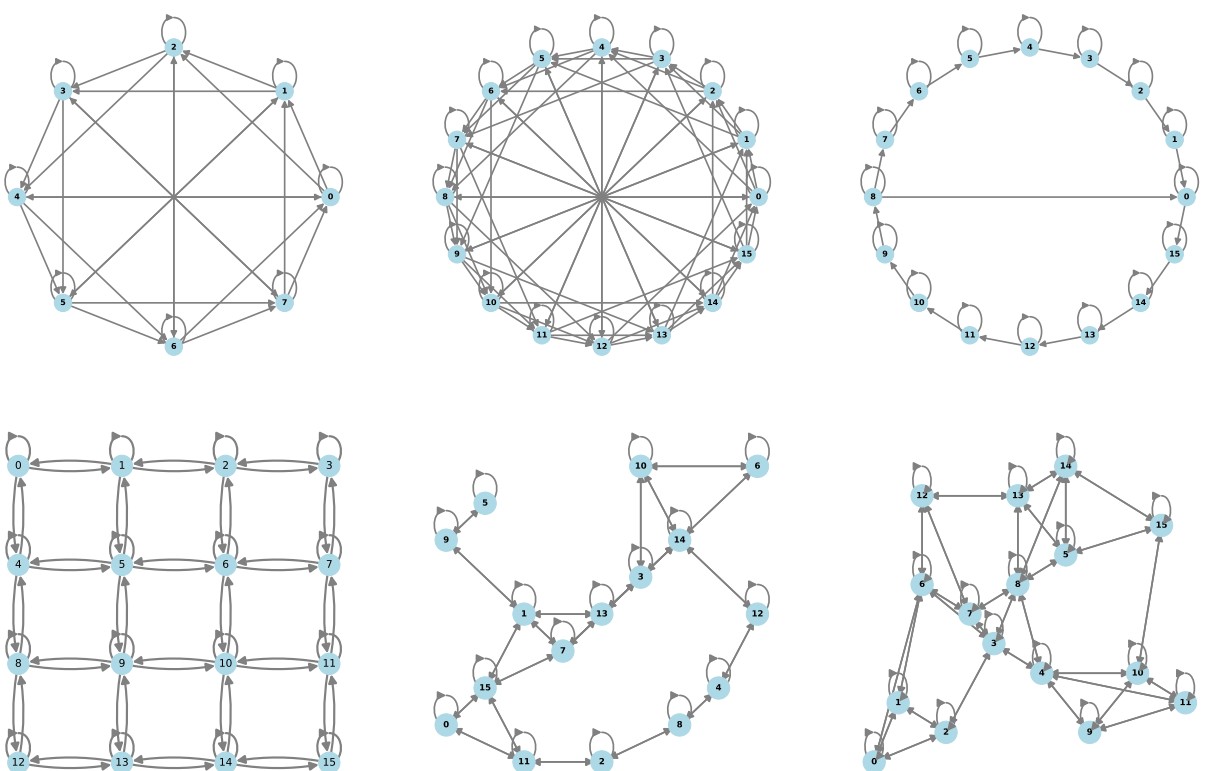

*Figure A1.* Directed exponential graphs with 8 and 16 nodes, and a directed ring graph (top three); an undirected grid graph, an undirected geometric graph, and an undirected nearest neighbor graph (buttom three).

Here we list the network metrics of the graphs above:

- Directed exponential graph with 8 nodes: $\beta_A = 0.5$, $\kappa_A = 1$.

- Directed exponential graph with 16 nodes: $\beta_A = 0.6$, $\kappa_A = 1$.

- Directed ring graph with 16 nodes: $\beta_A = 0.924$, $\kappa_A = 2$.

- Undirected grid graph with 16 nodes: $\beta_A = 0.887$, $\kappa_A = 6.25$.

- Undirected geometric graph with 16 nodes: $\beta_A = 0.916$, $\kappa_A = 3$.

- Undirected nearest neighbor graph with 16 nodes: $\beta_A = 0.907$, $\kappa_A = 1.74$.

## E.2. Synthetic Data Experiment

We conduct numerical experiments on a synthetic decentralized learning problem with non-convex regularization. The optimization function is given by:

$$f(x) = \frac{1}{n} \sum_{i=1}^{n} f_i(x), \ x \in \mathbb{R}^d$$

where

$$f_{i,L_i}(x) = \frac{1}{L_i} \sum_{l=1}^{L_i} \ln\left(1 + \exp(-y_{i,l} h_{i,l}^\top x)\right) + \rho \sum_{j=1}^{d} \frac{[x]_j^2}{1 + [x]_j^2}, \quad L_i \equiv M = L_{\text{Local}}/n$$

**Data Synthesis Process:**

1. **Global Data Generation:**

   - Generate optimal parameters: $x_{\text{opt}} \sim \mathcal{N}(0, I_d)$
   - Create global feature matrix $H \in \mathbb{R}^{L_{\text{total}} \times d}$ with $h_{l,j} \sim \mathcal{N}(0,1)$
   - Compute labels $Y \in \mathbb{R}^{L_{\text{total}}}$ with $y_l \in \{-1, +1\}$ via randomized thresholding:

   $$y_l = \begin{cases} 1 & \text{if } 1/z_l > 1 + \exp(-h_l^\top x_{\text{opt}}) \\ -1 & \text{otherwise} \end{cases}, \quad z_l \sim \mathcal{U}(0,1)$$

2. **Data Distribution:**

   - Partition $H$ and $L$ across $n$ nodes using:

   $$H^{(i)} = H[iM : (i+1)M, :], \quad Y^{(i)} = Y[iM : (i+1)M], \quad M = L_{\text{total}}/n$$

   where $L_{\text{total}}$ must be divisible by $n$

3. **Local Initial Points:**
   $$x_i^\star = x_{\text{opt}} + \varepsilon_i, \quad \varepsilon_i \sim \mathcal{N}(0, \sigma_h^2 I_d), \quad \sigma_h = 10$$

**Gradient Computation:** At each iteration, each node $i$ independently computes its stochastic gradient by randomly sampling a minibatch of size $B$ from its local dataset of size $L_i = L_{\text{total}}/n$. The gradient computation consists of two components:

$$\nabla f_{i,B}(x) = \underbrace{-\frac{1}{B} \sum_{b=1}^{B} \frac{y_{i,b} h_{i,b}}{1 + \exp(y_{i,b} h_{i,b}^\top x)}}_{\text{Logistic Loss Gradient}} + \underbrace{\rho \sum_{j=1}^{d} \frac{2[x]_j}{(1 + [x]_j^2)^2}}_{\text{Non-convex Regularization}} \tag{38}$$

where the minibatch $\{h_{i,b}, y_{i,b}\}_{b=1}^{B}$ is drawn uniformly from the $L_i$ local samples without replacement. Notice that the gradient on each node $i$ is computed on the local parameter $x_i \in \mathbb{R}^d$.

**Implementation Details:**

- Global dataset size $L_{\text{total}} = 204800$, batch size $B = 200$

- Dimension $d = 10$, regularization $\rho = 0.01$

- Node configurations: $n \in \{1, 2, 8, 16, 128, 512\}$

Notice that $L_{\text{total}} = 204800 = 512 \cdot 200 \cdot 2 = n_{\max} \cdot B \cdot 2$.

**Evaluation Metric:** We track $\|n^{-1} \mathbb{1}_n^\top \nabla f(\mathbf{x}^{(k)})\|$ as gradient norm, where

$$n^{-1} \mathbb{1}_n^\top \nabla f(\mathbf{x}^{(k)}) = \frac{1}{n} \sum_{i=1}^{n} \nabla f_{i,M}(\boldsymbol{x}_i^{(k)}), \ M = L_{\text{Total}}/n.$$

The experimental results shown in our plots represent the average performance across 20 independent repetitions, where each trial is conducted with the fixed random seed $= 42$ for reproducibility.

### E.3. Neural Network Experiment

Firstly, we employ a four-layer fully connected neural network for MNIST classification. The experiments are conducted on four distinct network topologies, each consisting of 16 nodes: a directed ring graph, an undirected grid graph, a geometric graph, and a nearest neighbor graph, as illustrated in Figure A1.

The MNIST training dataset, comprising 60,000 images, is evenly distributed across $n = 16$ nodes, each node maintaining an independent instance of the neural network model. At the end of each batch (that is, communication round), we record key performance metrics, including loss and accuracy. The communication is then executed based on the predefined network topology.

The complete experiment results are shown in A2. The accuracy cannot reach 100% because we are using a constant learning rate and the network is very simple.

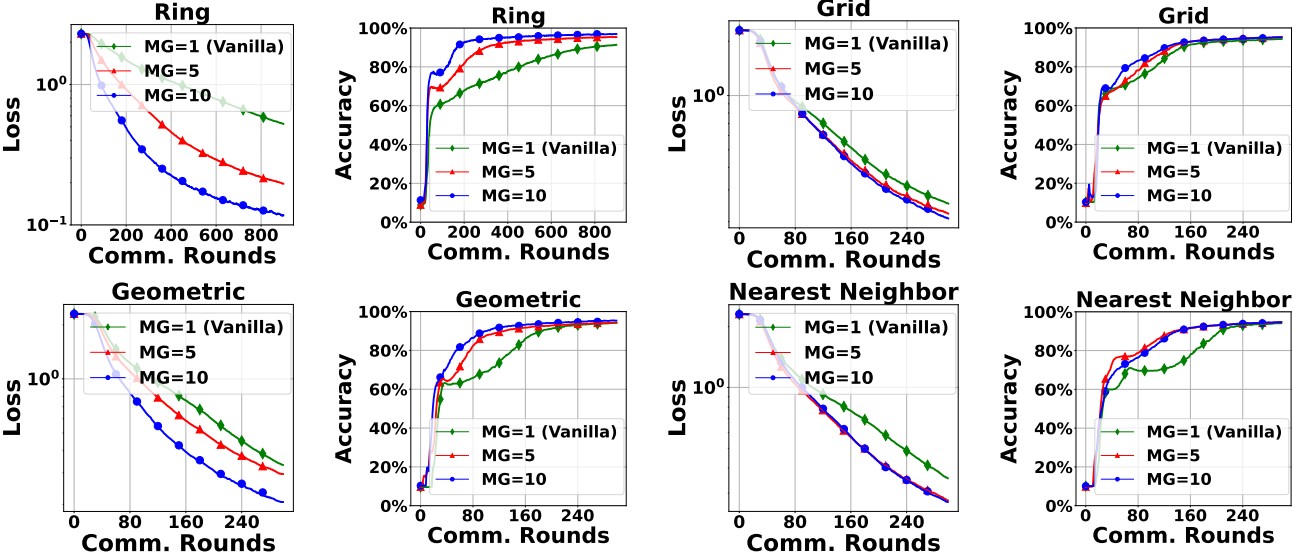

*Figure A2.* Training loss and test accuracy for models trained by MG-PULL-DIAG-GT and PULL-DIAG-GT on MNIST dataset. MG-PULL-DIAG-GT outperforms on all topologies.

Secondly, we employ a ResNet18 for CIFAR10 classification. The experiments are conducted on four distinct network topologies, each consisting of 16 nodes: a directed ring graph, an undirected grid graph, a geometric graph, and a nearest neighbor graph, as illustrated in Figure A1.

The experimental results are as follows:

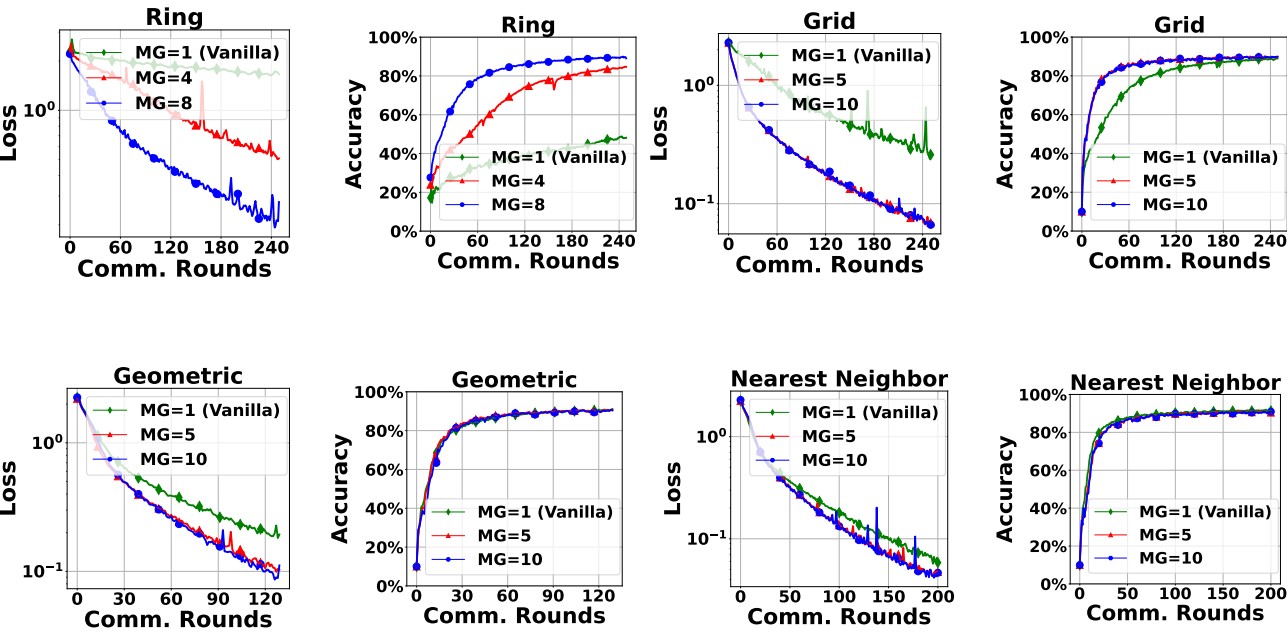

*Figure A3.* Training loss and test accuracy for models trained by MG-PULL-DIAG-GT and PULL-DIAG-GT on CIFAR-10 dataset. MG-PULL-DIAG-GT outperforms on all topologies.

Lastly, we provide an experiment on the heterogeneous MNIST dataset where each node possesses a unique data distribution, see Figure A4.

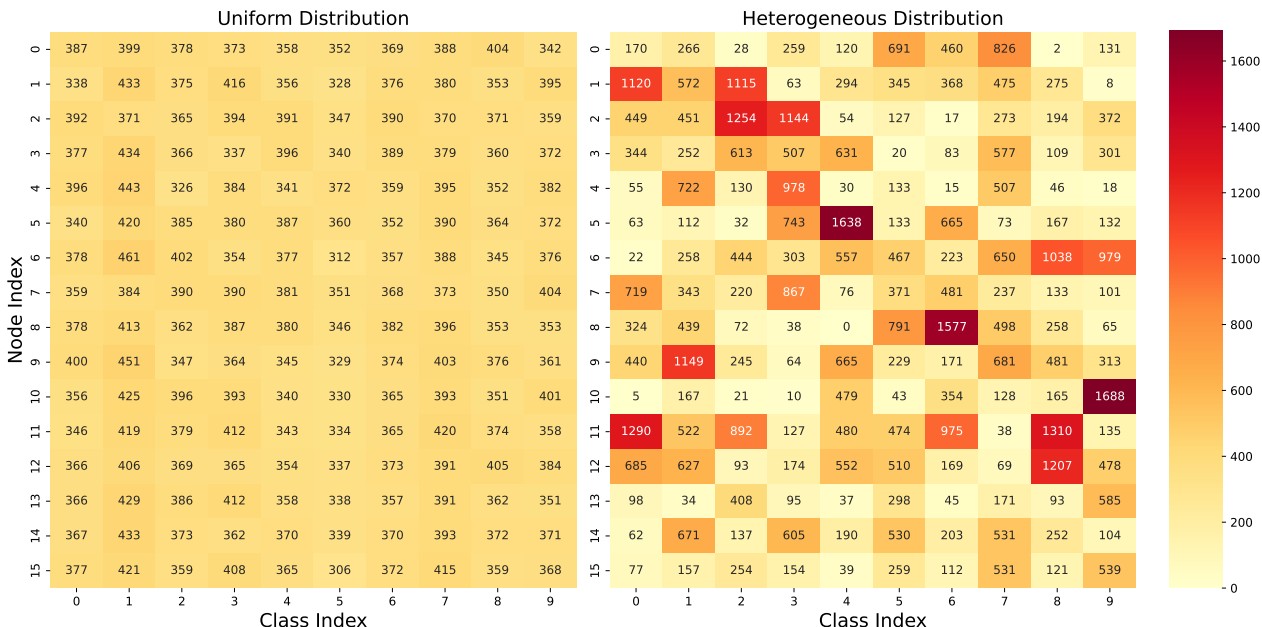

*Figure A4.* Left: Uniform MNIST data distribution used in the experiments in Section 6. Right: Heterogeneous MNIST data distribution used in the additional experiment shown in Figure A5.

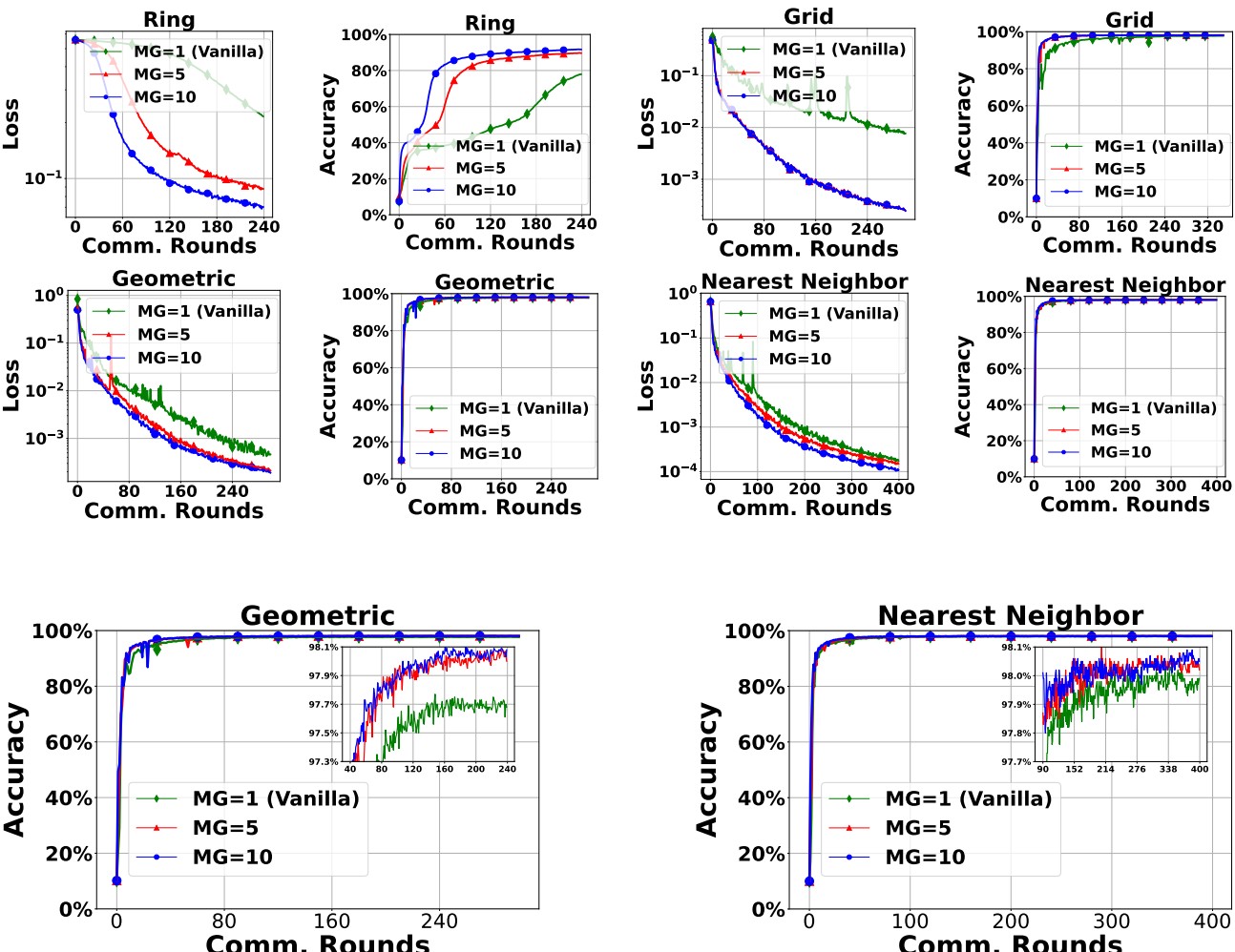

*Figure A5.* Training loss and test accuracy of a 4-layer neural network trained using MG-Pull-Diag-GT and vanilla Pull-Diag-GT on the MNIST dataset. The data is distributed in a heterogeneous manner (see Figure A4).

## E.4. Learning Rates

In Figure 2:

- Exponential graph of $n$ nodes:
$$n\alpha_n = 0.512.$$

- Ring graph of $n$ nodes:
$$n\alpha = 0.002$$

In Figure 3 and 4:

- For ring graph with $m$ times of multiple gossips, the learning rate $\alpha_m$ satisfies:
$$\alpha_1 = 0.005, \alpha_5 = 0.01, \alpha_{10} = 0.02.$$

.

- Grid graph:
$$\alpha_1 = 0.02, \alpha_5 = 0.03, \alpha_{10} = 0.03.$$

- Geometric graph:
$$\alpha_1 = 0.02, \alpha_5 = 0.02, \alpha_{10} = 0.03.$$

- Nearest neighbor graph:
$$\alpha_1 = 0.02, \alpha_5 = 0.02, \alpha_{10} = 0.02.$$

Here, the learning rates are chosen in a non-decreasing order, as the MG mechanism typically permits a wider range of stable learning rates.

