# OpenReview forum: "Achieving Linear Speedup and Near-Optimal Complexity for Decentralized Optimization over Row-stochastic Networks"
_ICML.cc/2025/Conference — ICML 2025 spotlightposter_

### Official Review · Reviewer_Si9i · 2025-03-12

**Overall Recommendation:** 3

**Summary:**

This paper studied the decentralized optimization problem where the mixing matrix is row-stochastic. This paper first derived the lower bound. Then, this paper analyzed PULL-DIAG-GT, showing that PULL-DIAG-GT requires an additional assumption, Assumption 4, to converge to the stationary point. Finally, this paper proposed a novel method, MG-PULL-DIAG-GT, which uses gradient accumulation and multiple gossip averaging. Then, this paper showed that MG-PULL-DIAG-GT can achieve almost the same convergence rate as the lower bound.

**Claims And Evidence:**

Yes

**Essential References Not Discussed:**

The related papers seem to be well discussed in this paper.

**Experimental Designs Or Analyses:**

See the comments in the suggestion.

**Methods And Evaluation Criteria:**

Yes

**Other Comments Or Suggestions:**

* Could you please describe how the authors tuned hyperparameters, e.g., learning rate, in Sec. 6? This information is very important for reproducing the experimental results.
* The authors used the exponential graph, grid, and ring as the network topology, while all of these graphs have a doubly stochastic mixing matrix. Could you please show some additional experimental results with a graph whose mixing matrix is not doubly stochastic?
* If the mixing matrix is doubly-stochastic, we need to use accelerated gossip averaging to achieve the optimal convergence rate [1,2]. The proposed method seems to achieve the lower bound without using this acceleration. Can the authors explain why acceleration is impossible (or unnecessary) when the mixing matrix is row-stochastic?


### Reference
[1] Tian, Y., Scutari, G., Cao, T., and Gasnikov, A. Acceleration in distributed optimization under similarity. In ICML 2022.

[2] Yuan, K., Huang, X., Chen, Y., Zhang, X., Zhang, Y., and Pan, P. Revisiting optimal convergence rate for smooth and non-convex stochastic decentralized optimization. In NeurIPS 2022

**Other Strengths And Weaknesses:**

N/A

**Questions For Authors:**

N/A

**Relation To Broader Scientific Literature:**

The proposed method seems to be a combination of existing techniques, gradient accumulation, multiple gossip averaging, and gradient tracking. However, deriving the lower bound of the convergence rate and showing that the proposed method can achieve almost the same rate as the lower bound are solid contributions to the research community.

**Theoretical Claims:**

No

---

> ### Author Rebuttal · Authors · 2025-03-31
>
> We sincerely thank the reviewer for their insightful feedback and constructive comments. Below, we address each point in detail.
>
> **Q1:** Could you please describe how the authors tuned hyperparameters, e.g., learning rate, in Sec. 6? This information is very important for reproducing the experimental results.
>
> **A1:** Thank you for your reminder. We will briefly describe how we tuned the learning rates and the logics behind it.
> - Figure 2: For exponential graph of $n$ nodes, the learning rate $\alpha_n$ satisfies $$\alpha_n\times n=0.0512.$$ For ring graph of $n$ nodes, the learning rate $\alpha_n$ satisfies $$\alpha_n\times n=0.002.$$ We set $n\times \alpha$ as a constant because, as shown in Appendix (lines 1268-1271), we have the bound
> $$\frac{1}{K+1}\sum_{k=0}^K \mathbb{E}[\|\nabla f(w^{(k)})\|^2] \le \frac{12\Delta}{ n\alpha(K+1)}+16\alpha L\sigma^2+  \frac{L\Delta}{K+1}.$$
> This implies that $n\alpha$ effectively controls the _descent rate_. By fixing $n\alpha$, we ensure that all curves decay at the same rate and only differ in their final noise-dominated error level.
>
> - Figure 3: For ring graph with $m$ times of multiple gossips,  the learning rate $\alpha_m$ satisfies: $$\alpha_1=0.005,\alpha_5=0.01,\alpha_{10}=0.02.$$
> For grid graph with $m$ times of multiple gossips,  the learning rate $\alpha_m$ satisfies: $$\alpha_1=0.02,\alpha_5=0.03,\alpha_{10}=0.03.$$
> For geometric graph with $m$ times of multiple gossips,  the learning rate $\alpha_m$ satisfies: $$\alpha_1=0.02,\alpha_5=0.02,\alpha_{10}=0.03.$$
> For nearest neighbor graph with $m$ times of multiple gossips,  the learning rate $\alpha_m$ satisfies: $$\alpha_1=0.02,\alpha_5=0.02,\alpha_{10}=0.02.$$
> Here, the learning rates are chosen in a non-decreasing order, as the MG mechanism typically permits a wider range of stable learning rates.
>
> We will include these in Appendix E in a future version.
>
> **Q2:** The authors used exponential graph, grid, and ring as the network topology, all of these graphs have a doubly stochastic mixing matrix. Could you please show some additional experimental results with a graph whose mixing matrix is not doubly stochastic?
>
> **A2:** Thank you for the question. While these graphs can support doubly stochastic matrices, we generate **row-stochastic** matrices over them, not doubly stochastic. We construct these matrices using the Metropolis rule (Nedic et al., 2018), where
> $$
> a_{ij} = \begin{cases}
> \frac{1}{1 + d_j^{\text{in}}}, & \text{if } (i \to j) \text{ exists} \\\\
> 0, & \text{otherwise}
> \end{cases}
> $$
> This typically yields a row-stochastic matrix. Only in the special case of regular graphs does the Metropolis rule produce a doubly stochastic matrix. We will emphasize that we used row-stochastic matrices in a future version.
>
> On the other hand, the row-stochasticity of these matrices can be verified by examining $\kappa_A$ listed in Appendix E (lines 1413–1421). Recall that $$\kappa_A = 1, \quad  \text{if and only if } A \text{ is doubly stochastic.}$$  For graphs ring, grid, geometric, and nearest neighbor, we have $\kappa_A > 1$, confirming that the mixing matrices are not doubly stochastic.
>
> **Q3:**  Can the authors explain why acceleration is impossible (or unnecessary) when the mixing matrix is row-stochastic?
>
> **A3:** Thank you for your insightful question. We will cite [1,2] and carefully discuss them in related works.
>
> - In [1,2], accelerated gossip relies on the mixing matrix being **doubly-stochastic and symmetric (or having real spectrum)**. This structure is crucial: after each power iteration (e.g., $A \to A^2$), the eigenvalues remain real and lie on the positive real axis. This allows a spectral shift (e.g., via Chebyshev polynomials) to center the spectrum around zero, reducing the spectral radius and enabling acceleration. In contrast, our setting involves **row-stochastic and generally non-symmetric** mixing matrices, for which the eigenvalues are complex and not aligned along the real axis. Without symmetry or spectral knowledge, Chebyshev-type acceleration becomes ineffective or even impossible.
>
> - Another reason lies in the network dependence on $1 - \beta$. In doubly-stochastic scenarios, the optimal dependence is $1/\sqrt{1 - \beta}$, which requires Chebyshev acceleration to achieve. In contrast, for row-stochastic matrices, the optimal dependence is $1/(1 - \beta)$, which is worse than in the doubly-stochastic case. Such $1/(1 - \beta)$ dependence can be attained without Chebyshev acceleration.
>
> **Other Comments:**
>
> While our proposed optimal algorithm combines existing algorithmic components, the **analysis technique is original** and departs from prior work in several key aspects. In particular, our work is the **first to handle "inexact descent"** in the nonconvex setting, whereas existing analyses focus on exact descent methods. We provide a detailed explanation of this distinction in our response to Reviewer zni5.

---

### Official Review · Reviewer_zni5 · 2025-03-12

**Overall Recommendation:** 3

**Summary:**

A key challenge in decentralized optimization is determining the optimal convergence rate and designing algorithms to achieve it. While this problem has been extensively addressed for doubly-stochastic and column-stochastic mixing matrices,
the row-stochastic scenario remains unexplored.
This paper bridges this gap by introducing effective metrics to capture the influence of row-stochastic mixing matrices and establishing the
first convergence lower bound for decentralized
learning over row-stochastic networks.

**Claims And Evidence:**

Yes

**Essential References Not Discussed:**

No

**Experimental Designs Or Analyses:**

Yes

**Methods And Evaluation Criteria:**

Yes

**Other Comments Or Suggestions:**

No

**Other Strengths And Weaknesses:**

Strength: This paper  introduces effective metrics to capture the influence of row-stochastic mixing matrices and establishing the
first convergence lower bound for decentralized
learning over row-stochastic networks.
Weakness: The techniques used in this paper are mainly from ``Towards better understanding the influence of directed networks
on decentralized stochastic optimization''.

**Questions For Authors:**

No

**Relation To Broader Scientific Literature:**

A good extension of previous work.

**Theoretical Claims:**

Yes

---

> ### Author Rebuttal · Authors · 2025-03-31
>
> **Reviewer's Comment:**
>
> Weakness: The techniques used in this paper are mainly from ''Towards better understanding the influence of directed networks on decentralized stochastic optimization''. (Liang et al. (2023))
>
> **Authors' Response:**
>
> We sincerely thank the reviewer for their insightful feedback and constructive comments.
>
> While our work is inspired by Liang et al. (2023), it differs significantly in both **problem setting** and **analytical techniques**. We encountered many new challenges and we developed original techniques for each of them,  listed as follows:
>
> **1. Challenges in Consensus Protocols.**
>
> - Liang's work studied Push-Sum protocol. In Push-Sum protocol, for a column-stochastic matrix $B$, its Perron vector $\pi_B$ is estimated by  $v^{(k)}:=n^{-1}B^k 1_n$. Due to the **linear** relationship $v^{(k+1)}=Bv^{(k)}$, the estimate from Push-Sum is consistent and enjoys good monotone properties (Lemma 2.1 in Liang et al. (2023)).
>
> - **Our work** studied Pull-Diag protocol. In Pull-Diag protocol, for a row-stochastic matrix $A$, the Perron vector $\pi_A$ can only be estimated by $d^{(k)}:=\rm{Diag}(A^k)$. Note that $d^{(k+1)}$ is **nonlinear** corresponding to $d^{(k)}$, we need to propose a novel analysis to study the property of Pull-Diag. See our Lemma 12.
>
> **2. Challenges in Obtaining Linear Speedup (Inexact Descent).**
>
> - Liang's work studied Push-Diging algorithm, which is an **exact** descent algorithm because the update follows (after proper projection):
> $$\hat{x}^{(k+1)}=\hat{x}^{(k)}-\gamma n\bar{g}^{(k)},$$
> where $\hat{x}^{(k)}$ is the projected parameter and $\bar{g}^{(k)}$ is the average of local gradients. This resembles **centralized SGD**, making the linear speedup result easy to be obtained.
>
> - **Our work** studied Pull-Diag-GT, which is an **inexact** descent algorithm because the update follows
> $$\hat{x}^{(k+1)}=\hat{x}^{(k)}-\gamma \pi_A^\top y^{(k)}.$$
> Compared to exact descent, we have an extra gap: $\pi_A^\top y^{(k)}-n\bar{g}^{(k)}$. This term, called **descent deviation**, typically prevents us from obtaining a linear speedup. Prior to our work, this term can only be dealt under the strongly-convex and deterministic setting. Our work is the **first** to rigorously handle such descent deviation in the nonconvex, $L$-smooth and stochastic setting. Lemmas 3, 4, 6, and 13 provide the tools for bounding and controlling this term. These lemmas can also be applied on similar inexact descent problems.
>
> **3. Challenges in Adapted Gradient Tracking.**
>
> - Liang's work studied standard gradient tracking, which is
> $$y^{(k+1)}=By^{(k)}+g^{(k+1)}-g^{(k)}.$$
> In standard gradient tracking, $g^{(k+1)}-g^{(k)}$ can be easily bounded using the $L$-smoothness property. This also enables a straightforward estimate for  $y^{(k)}$.
>
> - **Our work** studied adapted gradient tracking, which is
> $$y^{(k+1)}=Ay^{(k)}+D_{k+1}^{-1}g^{(k+1)}-D_k^{-1}g^{(k)}.$$
> The different coefficients $D_{k+1}$ and $D_k$ make it highly nontrivial to apply $L$-smoothness property. Our Lemma 4 provided valuable insights on how to address this problem.
>
> **4. Challenges Raised from Inversion of Small Values.**
>
> - In Push-Sum protocol studied by Liang et al (2023), the estimate of Perron vector $v^{(k)}$ is naturally lower bounded by a $\kappa$-related constant. They do not need to worry about the numerical instability.
>
> - **Our work** studies Pull-Diag protocol, which uses inversion of diagonal entries to do corrections. However, these diagonal entries can be arbitrarily small or zero, even for fixed $\kappa_A$ and $\beta_A$. To avoid being divided by zero, we must have Assumption 4 to provide a lower bound for diagonal entries. Altough this is a weak assumption (See our response to Reviewer MwZe, 3rd problem in 'Other Comments Or Suggestions'), this will prevent us from attaining the lower bound.
> Fortunately, in Lemma 7, we discover that MG will guarantee a positive lower bound on diagonal entries. This allows us to remove Assumption 4 and to derive a clear convergence rate using only $\kappa_A$ and $\beta_A$. While the final conclusion matches the conclusion in Liang et al. (2023), the underlying logic is quite different.
>
> In summary, our analysis provides a more general framework capable of handling richer sources of inexactness in decentralized optimization. We developed original techniques which differs from Liang et al. (2023).
>
> The challenges and distinctions above have also been discussed in our paper, pages 5 and 6.  A further clarification will be included in Appendix due to page limits.

---

### Official Review · Reviewer_MwZe · 2025-03-24

**Overall Recommendation:** 3

**Summary:**

This paper presents a theoretical analysis of decentralized optimization with row-stochastic mixing matrices. It is the first to establish a lower bound for convergence. Gradient tracking-based algorithms are shown to achieve linear speedup in convergence under an additional assumption. To overcome this limitation and attain a near-optimal convergence rate, the authors propose a new algorithm that incorporates multiple gossips. Experimental results validate the theoretical findings.

**Claims And Evidence:**

The claims in the theory part are clear, with step-by-step sketch of the proofs, which are easy to understand. However, the experiments are not as convincing as the math derivations.

**Essential References Not Discussed:**

I suggest adding more recent works. For example, in Section 1, most papers look like before pandemic. In addition, I would like to see more practical evidences of analyzing directed graph. E.g., there is one paper for differences in node power ranges, and what about channel disruptions?

**Experimental Designs Or Analyses:**

Experiments look traditional. I have a few questions on the design:
1. Is it possible to showcase the MG component in a larger real dataset?
2. The graph structure is either un-directed or directed with good structure. How will the graph topology change the results? For example, a random directed graph, and/or even with high $\kappa_A$?
3. How is the MNIST data distributed. Are they homogenous (uniform)?
4. Is there any explanation on the plateau-like stage of the accuracy in the neural net experiments?

**Methods And Evaluation Criteria:**

Yes, techniques used in this paper, as well as the performance evaluation, are widely accepted in this field.

**Other Comments Or Suggestions:**

1. In Section 2.3, right before Section 3, the referred Figure 1 shows exponential rate of convergence. I would guess it is because it works on a strongly-convex problem. However, I did not find where the description of the setting locates (or I missed it). This might cause misunderstanding of the claim of convergence rate, as your assumptions are much milder than strong convexity.
2. I suggest explaining the abbreviation GT before using it in the abstract, similar to MG.
3. it could be better to discuss the limitation of Assumption 4, as it looks unnatural and it is hard to tell which scenarios satisfy or un-satisfy this assumption.
4. In Section 2 Assumption 1, the meaning of $(i,j)\in\mathcal{E}$ needs clarification, as this is a directed graph.

**Other Strengths And Weaknesses:**

No more comments here. Important questions or suggestions are **numbered** in the corresponding parts.

**Questions For Authors:**

1. This paper closely follows the approach of Liang et al. (2023), including the use of a specific metric, a similar multi-gossip (MG) strategy to address optimal convergence, and analogous forms of the resulting bounds. As a result, the theoretical originality here sounds unclear to me. To strengthen the contribution, it is important for the authors to clearly articulate the technical novelties and distinctions from the literature, beyond applying known ideas in a slightly different setting.

2. Would it be more accurate to describe the result as near-optimal, given the logarithmic gap between Theorem 3 and Theorem 1? I find the phrasing in the title of Section 5—“near-optimal rate”—both appropriate and reflective of the paper’s contributions.

**Relation To Broader Scientific Literature:**

This paper inherits ideas from Liang et al. (2023) with same metric definitions, similar MG remedy for the convergence, and analogous bounds. Therefore, the results in this paper sounds reasonable as everything looks like a transpose (to me). It could be better if the authors are able to emphasize the technical difference between their results and that literature work.

**Theoretical Claims:**

Yes, theoretical claims look good to me.

---

> ### Author Rebuttal · Authors · 2025-03-31
>
> We sincerely thank the reviewer for their insightful feedback and constructive comments. Below, we address each point in detail. **All newly added experiments can be found in the Rebuttal Experiment Sheet (RES)** https://anonymous.4open.science/r/ICML-2025-Rebuttal-Experiment-Sheet-B6C0/
>
> **Experimental Designs Or Analyses**
>
> 1. We add an experiment training ResNet-18 on CIFAR-10, using both vanilla Pull-Diag and its MG versions. Data is uniformly distributed. See Figure 1 in RES.
>
> 2. Both $\beta_A$ and $\kappa_A$ will affect the results. For most topologies (including random graphs), $\kappa_A$ is typically a small number and convergence rate is mainly affected by $\beta_A$. For special mixing matrices, as shown in our Proposition 8, $\kappa_A$ can be exponentially large. In this case, vanilla Pull-Diag-GT suffers a lot, while its MG version can achieve quick convergence and higher precision. We present experiments for large $\kappa_A$ in Figure 2 in RES.
>
> 3. Yes, MNIST data is uniformly distributed. For heterogeneous results, see Figures 3,4 in RES.
>
> 4. Since we use a constant learning rate and a simple 4-layer neural network, the test accuracy will not reach 100\% in the end.
>
> **Relation To Broader Scientific Literature**
>
> Our work addresses a series of new challenges with original techniques. **These challenges and techniques are detailed in our response to Reviewer zni5.** For reviewer's convenience, we also provide a brief summary as follows:
>
> i. **Consensus protocol: Pull-Diag is fundamentally different from Push-Sum.** Pull-Diag introduces stronger nonlinearities and disrupts the linear structure exploited in Push-Sum analysis. It is not a dual or transpose of Push-Sum, thus new analytical tools are needed. See Lemma 12.
>
> ii. **New error terms.** Liang et al. only handle consensus error. In our case, we must additionally control descent deviation, which arises from the interaction between Pull-Diag and gradient tracking. This term is new in nonconvex settings and is analyzed via Lemmas 2 and 3.
>
> iii. **Non-uniform gradient tracking**: In Push-DiGing, all gradients carry the same weight, allowing standard $L$-smoothness arguments. In our method (Eq. 5b), gradient coefficients vary, breaking this symmetry. We introduce Lemma 4 to handle this non-uniformity.
>
> iv. **Inversion of small values.** Pull-Diag involves inverting diagonal entries that can be arbitrarily small, making it hard to control bounds. In Lemma 7, we show that MG guarantees a positive lower bound on these values, allowing us to derive clear convergence rates in terms of $\kappa$ and $\beta$. While this matches a conclusion in Liang et al., the underlying logic is different.
>
> In a nutshell, our work provides new analytical tools to handle the unique challenges introduced by Pull-Diag and Row-Only settings. These tools can be extended to analyze other algorithms in similar settings.
>
> **Essential References Not Discussed**
>
> We will discuss connection failure (Yemini et al.,2022, Li et al 2024) and include more recent literature on Row-Only algorithms (Jeong and Kountouris, 2024, Nguyen et al. 2023, Xing et al., 2024).
>
> **Other Comments Or Suggestions**
>
> 1. The problem in Figure 1 is ''achieving average consensus''. Initially, every agent $i$ has vector $z_i$. Agents communicate with their neighbors and all want$\frac{1}{n}\sum_{i=1}^n z_i$ in the end. The $y$-axis denotes the distance between their current state and $\frac{1}{n}\sum_{i=1}^n z_i$. This can be seen as a strongly convex optimization problem. We plot Figure 1 to show the impact of $\beta_A$ and $\kappa_A$ on Pull-Diag, thereby justifying their use in characterizing row-stochastic matrices. However, we do not use Figure 1 to support our theoretical results in non-convex settings.
>
> 2. Our abstract will be modified as:
>
> ... deviation in the descent direction caused by the adapted gradient tracking (GT) and instability introduced by the PULL-DIAG protocol.
>
> 3. Under Assumption 1, an equivalent statement of Assumption 4 is: $A$ has a positive diagonal (every node has a self-loop). This equivalence can be derived using Lemma 7. We will include this in a future version.
>
> 4. $(i,j) \in E$ represents an edge from $i$ to $j$.
>
> **Questions For Authors**
>
> For Question 1,
> - Our work differs from Liang et al. (2023) in both setting and analysis. Please check our response to **Relation To Broader Scientific Literature** and a more detailed version can be found in our response to Reviewer zni5.
>
> - The key novelty is that, our analysis framework is more general. Liang et al. analyzes **exact** descent method, while our method addresses **inexact** descent with disturbacnes.  Detailed explanation of this distinction can also be found in our response to Reviewer zni5.
>
> - We have discussed this distinction in our paper pages 5 and 6, and a further clarification will be included in Appendix due to page limits.
>
> For Question 2,
> - Yes. We would use near-optimal in title in a future version.

---

### Decision · Program_Chairs · 2025-05-01

**Decision:**

Accept (spotlight poster)

**Comment:**

This paper aims at determining the optimal convergence rate for decentralized optimization over a row stochastic network and designing an algorithm to achieve it. For the former, new metrics are designed to capture the influence of row-stochastic mixing matrices. For the latter, a novel analytical framework and a multiple-step gossip protocol are combined so as to design an optimal algorithm. Although the proposed method is a combination of existing techniques, the reviewers agree that the contributions are solid. The authors are encouraged to further highlight the new features compared to the existing works. All reviewers unanimously recommend acceptance.